# Settling the Communication Complexity for Distributed Offline Reinforcement Learning

## Abstract

We study a novel setting in offline reinforcement learning (RL) where a number of distributed machines jointly cooperate to solve the problem but only one single round of communication is allowed and there is a budget constraint on the total number of information (in terms of bits) that each machine can send out. For value function prediction in contextual bandits, and both episodic and non-episodic MDPs, we establish information-theoretic lower bounds on the minimax risk for distributed statistical estimators; this reveals the minimum amount of communication required by any offline RL algorithms. Specifically, for contextual bandits, we show that the number of bits must scale at least as $\Omega(AC)$ to match the centralised minimax optimal rate, where $A$ is the number of actions and $C$ is the context dimension; meanwhile, we reach similar results in the MDP settings. Furthermore, we develop learning algorithms based on least-squares estimates and Monte-Carlo return estimates and provide a sharp analysis showing that they can achieve optimal risk up to logarithmic factors. Additionally, we also show that temporal difference is unable to efficiently utilise information from all available devices under the single-round communication setting due to the initial bias of this method. To our best knowledge, this paper presents the first minimax lower bounds for distributed offline RL problems.

## 1 Introduction

In this paper, we study a problem setting where each device can only send one message to the central server and the number of information in terms of bits is constrained by a *communication budget*. After that one transmission, no further communication is allowed; this setting models the situation where each device has a limited connection to the central server. Examples of this problem are ubiquitous in the field of electronic wearables and IoT devices, which are often low-powered and have limited access to internet. In such cases, connection with global server is rare and can be effectively modelled as a one-shot. If one wants to use Reinforcement learning approaches on such devices, each device would require access to large amounts of past experiences to efficiently solve the task. Fortunately, if one combines the data gathered on all devices in some region, within this collective experience we are likely to have explored enough of the environment to achieve acceptable performance. However, to utilise the data from other devices the RL algorithm needs to be able to communicate efficiently with other devices and learn from this collective experience in an offline manner. It is expected that enforcing restrictions on the available communication will decrease the overall performance for a given task. Yet, the research questions of how does the best possible performance depend on the communication budget, and how does existing offline RL algorithms perform have never been answered.

In this paper, we attempt to answer the above questions by studying distributed offline RL problems. Our proof methods are based on reducing RL problems to statistical inference problems. After such reduction is done we utilise framework developed by Zhang et al. (2013), which ties minimax risk to mutual information of parameter and message sent. Intuitively, if a message carries little information about a certain parameter, it will be difficult for the central server to perform accurate inference based on that message only. We thus bound the mutual information to derive a lower bound on the risk any one-shot communication algorithm must suffer. Under our setting, we start with the classical multi-armed bandits problem in the contextual

| Problems | Algorithms | Optimal Risk | Bits Communicated |
|---|---|---|---|
| Parameter learning in linear contextual bandits | LSE (Algorithm 1) | $O(\frac{ACR^2}{mn\lambda})$ | $\Theta(AC \log \frac{mn\lambda}{R})$ |
| | Lower Bound | $\Omega(\frac{ACR^2}{mn\lambda})$ | $\Omega(AC)$ |
| Value function prediction (Episodic MDP) | MC LSE (Algorithm 2) | $O(\frac{CH^2R^2}{mSE\lambda})$ | $\Theta(C \log \frac{mSE\lambda}{HR})$ |
| | Lower Bound | $\Omega(\frac{CH^2R^2}{SEm\lambda})$ | $\Omega(\frac{C}{\log m})$ |
| Value function prediction (Non-Episodic MDP) | TD (Algorithm 3) | $O(\frac{\nu}{1+(1-\gamma)^2 n})$ | $\Theta(C \log \frac{C}{\nu})$ |
| | Lower Bound | $\Omega(\frac{CR^2}{(1-\gamma)^2 mSn\pi_{\max}\lambda})$ | $\Omega(\frac{C}{\log m})$ |

Table 1: Summary of main results. $A$ denotes the cardinality action space, $S$ card. of state space, $C$ is the dimensionality of context/feature vectors, $m$ is the number of cooperating machines, $n$ is the number of samples, $E$ number of episodes, $\lambda$ is the smallest rescaled eigenvalue of the feature/context covariance matrix, $H$ is the horizon length, $\gamma$ denotes the discount factor. For the stationary distribution of states visited in non-episodic MDP $\pi$, we assume that $\pi_{\max} \geq \pi(s) \geq \pi_{\min}$. We define $\nu = \max\{\frac{R^2}{S\pi_{\min}\lambda m}, \|\boldsymbol{\theta} - \frac{1}{m}\sum_{i=1}^{m}\hat{\boldsymbol{\theta}}_0^i\|_2^2\}$, where the second argument of max operator is the initial bias of TD method.

case, where we derive a lower bound on the risk of $\frac{ACR^2}{mn\lambda}$ when the communication budget scales at least as $AC$, where $A$ and $C$ are number of actions and dimensions of context respectively, $R$ is the maximum reward, $m$ and $n$ are numbers of machines and samples for each action and $\lambda$ is the rescaled smallest eigenvalue of the covariance matrix of context/feature vectors. We also present Algorithm 1 achieving optimal risk with communication budget being optimal up to logarithmic factors. Existing analyses of communication in multi-armed bandits problems (Wang et al. (2019), Huang et al. (2021)) define the communication cost as the number of values transmitted rather than the number of bits, which unfortunately ignores the fact that real communication can only occur with finite precision.

Apart from the bandits problem, we also analyse problem settings for Markov Decision Processes (MDPs) and develop lower bounds for both episodic and non-episodic settings. To our best knowledge, we present the first minimax lower bounds for distributed solving of MDPs. In the episodic setting, we develop a lower bound of $\frac{CH^2R^2}{mSE\lambda}$ of risk with $\frac{C}{\log m}$ communication budget, where $H$ is horizon length, $S$ is the cardinality of state space and $E$ is the number of episodes we have for each state. We also prove that performing distributed Monte-Carlo return estimates achieve optimal risk with communication budget being optimal up to logarithmic factors. Additionally, we extend our results to the non-episodic cases and develop a lower bound of $\frac{CR^2}{(1-\gamma)^2 mSn\pi_{\max}\lambda}$ with communication budget $\frac{C}{\log m}$. Meanwhile, we also study the worst-case risk of distributed temporal difference in this setting and show that with only one communication round, the algorithm **cannot** efficiently utilise data stored on all machines due to its initial bias, i.e., the difference between the average initial estimate and the true parameter value $\|\boldsymbol{\theta} - \frac{1}{m}\sum_{i=1}^{m}\boldsymbol{\theta}_0^i\|_2$, which does not decrease with the number of machines.

We summarise our theoretical results in Table 1 and our contributions are listed as follows:

- We present information-theoretic lower bounds for distributed offline linear contextual bandits and linear value function prediction in Markov Decision Processes. Our lower bounds scale with the allowed communication budget $B$. To the best of the authors' knowledge, these are the first lower bounds in distributed reinforcement learning other than for bandit problems.

- We prove that a distributed version of the least-square estimate achieves optimal risk for the contextual bandit problem and a distributed version of Monte-Carlo return estimates achieves optimal risk in episodic MDP. We show that both of these algorithms have communication budgets optimal up to logarithmic factors.

- We show that the performance of distributed temporal difference in one communication round setting does not improve when data from more machines is used if the initial bias is large

## 2 Problem Formulation

We assume there are $m$ processing centres, and the $i$th centre stores gameplay history $h^i$ of an agent inter-acting with a particular environment within a framework of a Markov Decision Process. The distribution of gameplay histories $P(h)$ belongs to some wider family of distributions $\mathcal{P}$, which is taken to be a set of all possible distributions that might have generated gameplay given that they satisfy assumptions of the problem. We assume there is some parameter of interest $\boldsymbol{\theta} : \mathcal{P} \rightarrow \Theta$ embedded within that problem, which might be a property of solely the environment or a result of the agent's actions. All centres are supposed to cooperate to jointly obtain an estimate $\hat{\boldsymbol{\theta}}$, closest to the true value $\boldsymbol{\theta}$. Based on its gameplay history $h^i$, the $i$th centre can send a message $Y^i$ to the main processing centre (arbitrary chosen), according to some communication protocol $\Pi$. There is only one communication round allowed, and the main centre cannot send any message back to individual centres. After receiving messages from all centres $Y^1, \ldots, Y^m$ the main centre outputs an estimate $\hat{\boldsymbol{\theta}}(Y^1, \ldots, Y^m)$. We define the worst-case risk of that estimate in Definition 2.1.

**Definition 2.1** *For a problem of estimating a parameter $\boldsymbol{\theta} : \mathcal{P} \rightarrow \Theta$, we define the worst-case risk of an estimator $\hat{\boldsymbol{\theta}}$ under communication protocol $\Pi$ as*

$$W(\boldsymbol{\theta}, \mathcal{P}, \hat{\boldsymbol{\theta}}, \Pi) = \sup_{P \in \mathcal{P}} \mathbb{E}\big[||\hat{\boldsymbol{\theta}}(Y^{1:m}) - \boldsymbol{\theta}(P)||_2^2\big]$$

*where the expectation is taken over possible gameplay $h^i$ histories generated by $P$ and messages $Y^{1:m}$ sent by the protocol $\Pi$ based on them.*

We now consider the case when each message $Y^i$ is constrained so that its length in bits $L^i$ is smaller than some communication budget $B$. We define $A(B)$ to be the family of communication protocols under which $\forall_{i \in \{1, \ldots, m\}} L^i \leq B$. We define the minimax risk as the best possible worst-case risk any algorithm can have as stated by Definition 2.2.

**Definition 2.2** *For a class of problems where gameplay history is generated by a distribution $p \in \mathcal{P}$ and we wish to estimate a parameter $\boldsymbol{\theta} : \mathcal{P} \rightarrow \Theta$, under a communication budget of $B$ we define the minimax risk to be:*

$$M(\boldsymbol{\theta}, \mathcal{P}, B) = \inf_{\Pi \in A(B)} \inf_{\hat{\theta}} W(\boldsymbol{\theta}, \mathcal{P}, \hat{\boldsymbol{\theta}}, \Pi) = \inf_{\Pi \in A(B)} \inf_{\hat{\theta}} \sup_{P \in \mathcal{P}} \mathbb{E}\big[||\hat{\boldsymbol{\theta}}(Y^{1:m}) - \boldsymbol{\theta}(P)||_2^2\big]$$

*where the second infimum is taken over all possible estimators.*

Notably, a similar definition was adopted by Zhang et al. (2013) in the setting of distributed statistical inference. The lower bounds we formulate are given in form of a hard instance of a problem. Formally, we show that there exists an instance such that any algorithm has a minimax risk at least equal to the bound we derive. We use a method described in Appendix A, which gives us a lower bound on minimax risk, provided we can obtain an upper bound of mutual information the messages sent by machines carry about the parameter $I(Y, V)$. To this goal, we rely on inequality stated in Appendix A.3, which requires us to construct a set such that if samples we receive fall into it, the likelihood ratio given different values of the parameter is bounded. If we can additionally bound the probability that the samples do not fall into this set, which in our proofs is done through the Hoeffding inequality, we obtain an upper bound on the information. Intuitively, if the likelihood ratio is small, it becomes hard to identify a parameter and if this also happens with large probability, then average information $I(Y, V)$ messages carry about the parameter must be small. We will now introduce the communication model assumed in this paper.

### 2.1 Information Transmitted and Quantisation Precision

In every algorithm we propose, we assume the following transmission model; we introduce quantisation levels separated by $P$, which we call the precision of quantisation. We assume the transmitted values are always within some pre-defined range $(V_{\min}, V_{\max})$, hence we divide it into $\frac{V_{\max} - V_{\min}}{P}$ levels. Each value gets quantised into the nearest level so that the difference between the original value and the level into which it was quantised is smallest. Under such a scheme, the number of bits transmitted is, therefore $B = \log\left(\frac{V_{\max} - V_{\min}}{P}\right)$. We now proceed with the analysis of the first class of the distributed offline RL problems.

# 3 Parameter estimation in contextual linear bandits

In a classic multi-armed bandit problem, the agent is faced with a number of slot machines and is allowed to pull the arm of only one of them at a given timestep. We can identify choosing each arm with choosing an action $a \in \mathcal{A}$, where $A = |\mathcal{A}|$ is the number of all arms. The reward obtained after each pull is stochastic and is sampled from a distribution that depends on the machine. We do not specify the distribution of the reward, but put a standard constraint on the maximum absolute value of rewards as stated by Assumption 3.1, restricting its support to $[-R_{\max}, R_{\max}]$.

**Assumption 3.1** *The maximum absolute value of the reward an agent can receive at each timestep $t$ is bounded by some constant $R_{max} > 0$, i.e. $|r_t| \le R_{max}$.*

Hence, the goal in such a task is to identify the arm with the highest average reward. In the contextual version of this problem, at each time step, the agent is also given a context vector $c_t \in \mathbb{R}^C$, which influences the distribution of the reward for each arm. Consequently, an arm that is optimal under one context need not be optimal under a different context. We explicitly assume a linear structure between the context vector and average reward for each arm, i.e. $\mathbb{E}[r_t|a_t] = c_t^T \theta_{a_t}$, where $\theta_a \in \mathbb{R}^C$ is the parameter vector for the arm associated with action $a$. We analyse the offline version of this problem where we have access to the gameplay history $h = \{c^l, a^l, r^l\}_{l=1}^N$, consisting of received context vectors, the arms agent have chosen in response to them and rewards obtained after pulling the arm. We assume that within gameplay history the agent has chosen each arm $n$ times so that $N = An$. We make a further assumption regarding the context vectors.

**Assumption 3.2** *Context vector for each sample is normalised, i.e. $\|c_i\|_2^2 \le 1$. If we form a matrix $X = (c_1, \ldots, c_n)$ for any number of context vectors $n$, then the smallest eigenvalue of the matrix $X^T X$ is $\eta_{min}$ such that $0 < \eta_{min} \le 1$ and $\eta_{min} = n\lambda_{min}$ for some constant $\lambda_{min}$.*

Restricting the smallest eigenvalue is necessary as otherwise, the parameter $\theta_a$ becomes unidentifiable, as large changes in it will produce small changes in the average reward. Also, in practical problems, eigenvalues of the covariance matrix naturally grow with the number of samples and the condition that $\eta_{\min} = n\lambda_{\min}$ has been adopted by similar analyses (Zhang et al. (2013)). We would like to obtain an estimator for $\theta = (\theta_1, \ldots, \theta_A)$, which is a concatenated vector consisting of all parameter vectors for each $a \in \mathcal{A}$. We now proceed with presenting the minimax lower bound for the estimation of $\theta$. We assume the context vectors machines receive are pre-specified and the only randomness is within the reward. We now present our first lower bound on minimax risk in Theorem 3.3 and sketch its proof, describing a hard instance of the contextual linear bandit problem.

**Theorem 3.3** *In a distributed offline linear contextual MAB problem with $A$ actions and context with dimensionality $C$ such that $CA > 12$ under assumptions 3.1 and 3.2, given $m$ processing centres each with $n \ge C$ samples for each arm, with each centre having communication budget $B \ge 1$, for any independent communication protocol, the minimax risk $M$ is lower bounded as follows:*

$$M \ge \Omega\left( \frac{ACR_{max}^2}{mn\lambda_{min}} \min\left\{ \max\left\{ \frac{AC}{B}, 1 \right\}, m \right\} \right)$$

**Proof 3.4** *(sketch): We present a sketch of proof here and defer the full proof to Appendix B.*
*We construct a hard instance of the problem and use it to derive a lower bound. For each arm, at each centre we set the $k$th component of context vector in $l$th sample to $c_k^l = \sqrt{\lambda_{min}n}\mathbb{1}_{k=l}$ for the first $C$ and we set context vectors for remaining $n - C$ samples to zero vectors. Although this construction might seem pathological at first glance, it satisfies Assumption 3.2 and simplifies the analysis greatly. Under our construction we choose the distribution of the reward $r_j^l$ for $l$th sample for $j$th arm to be $R_{max}$ with prob. $\frac{1}{2} + \frac{c_l^l \theta_j^l}{2R_{max}}$ and $-R_{max}$ otherwise. We can observe that the expected reward is $\mathbb{E}[r_j^l] = c_l^l \theta_{j,l} = (c^l)^T \theta_j$, which satisfies the problem formulation. The reward can also be rewritten as $r_j^l = (2p_j^l - 1)R_{max}$, where $p_j^l \sim Bernoulli(\frac{1}{2} + \frac{c_k^l \theta_j}{2R_{max}})$. Hence the data received can be just reduced to the Bernoulli variables $\{p_j^l\}_{l=1}^n$ for each arm $j$. We follow the method*

*described in Appendix A. We study the likelihood ratio of $p_j^l$ under $v_{j,k}$ and $v'_{j,k} = -v_{j,k}$ and show that it is bounded by $\exp\left\{\frac{17\sqrt{n\lambda_{min}}\delta v_k}{8R_{max}}\right\}$. We can thus satisfy the conditions of Lemma A.4, with $\alpha = \frac{17\sqrt{n\lambda_{min}}\delta v_k}{8R_{max}}$. We observe that since we haven't assumed anything about $p_j^l$ to derive this bound, we get that this bound holds with probability one and hence the second and third term in the bound resulting from Lemma A.4 are zero. For the Bernoulli distribution we can easily study the KL-divergence and together with Lemma A.2 we get that: $I(V,Y) \leq \frac{17}{8}\frac{\delta^2 mn\lambda_{min}}{R_{max}^2}\min\left\{\frac{17}{8}B, \frac{CA}{2}\right\}$. The rest of the proof consists of choosing such $\delta$ that produces the tightest bound.*

We observe that for the communication not be a bottleneck, we require a communication budget for each machine of at least $B > \Omega(AC)$. We also see that above this optimal threshold, increasing the communication budget does not decrease the lower bound because of the max operator. On the other hand, for small communication budgets, the min operator ensures that the performance cannot be worse than as if we only used data from one machine. Both of these results are intuitive and show that our bound is thigh with respect to the communication. Since the optimal communication budget scales with $A$, this lets us presume that we can tackle this problem by performing $A$ separate least-squares estimations. Inspired by that, we present Algorithm 1 and further show in Theorem 3.5 that it can match this lower bound up to logarithmic factors.

---

**Algorithm 1** (LSE) Distributed offline least-squares

---

**Data:** $\{c_a^l, r_a^l\}_{l=1}^n$ for $a \in \mathcal{A}$
On individual machines compute:
**for** $i \in \{1, \ldots, m\}$ **do**
    **for** $a \in A$ **do**
        $X_a \leftarrow (c_a^1, \ldots, c_a^n)^T$   $r_a \leftarrow (r_a^1, \ldots, r_a^n)^T$
        $\hat{\theta}_a^i \leftarrow (X_a^T X_a)^{-1} X_a^T r_a$
    **end**
    Quantise each component of $\hat{\theta}_a^{\,i}$ up to precision $P$ and send to central server.
**end**
At central server compute: $\hat{\theta}_a \leftarrow \frac{1}{m}\sum_{i=1}^m \hat{\theta}_a^i$ for $a \in \mathcal{A}$
**return** $\theta_a$ for each $a \in \mathcal{A}$

---

**Theorem 3.5** *Let us define $\theta = (\theta_1, \ldots, \theta_A)$ to be the concatenated parameter vector for all actions. For any value of $\theta$, Algorithm 1 using transmission with precision $P$ achieves a worst-case risk upper bounded as follows:*

$$W < O\left(AC\max\left\{\frac{R_{max}^2}{mn\lambda_{min}}, P\right\}\right)$$

**Proof 3.6** *(sketch): We present a sketch of proof here and defer the full proof to Appendix E. We start by assuming the transmission is lossless (i.e. the number of bits is infinite) and then study how the MSE changes when transmission introduces quantisation error. It is a well-known fact that for the least-Squares estimate, the bias is zero, hence using the bias-variance decomposition, we get that the bound on estimator's variance is also the bound on its MSE. Some algebraic manipulations allow us to show that $Var(\hat{\theta}_{a,k}^i) \leq \sum_{j=1}^C \sum_{l=1}^n \frac{Q_{j,k}^2 V_{j,l}^2}{\lambda_{min} n} Var(r_a^l)$. We now observe that because of Assumption 3.1, we can use Popoviciu inequality to obtain $Var(r_a^l) \leq R_{max}^2$. Substituting this back into the equation for MSE, we get $\mathbb{E}[(\theta - \hat{\theta})^2] \leq \frac{ACR_{max}^2}{mn\lambda_{min}}$. Using the fact that each component is quantised up to precision $P$, the max error introduced by quantisation is $ACP$. Combining that with the bound on MSE of lossless transmission, we get the statement of the Theorem.*

A direct conclusion of this result is that if we want the quantisation to not affect the worst-case performance of Algorithm 1, the precision $P$ needs to scale as $\Theta(\frac{R_{max}^2}{mn\lambda_{min}})$ and hence the number of transmitted bits $B$ must scale as $\Theta(AC \log \frac{mn\lambda_{min}}{R_{max}^2})$. While enjoying the simplicity of analysis, the contextual bandit problem does not

take into account that the current actions of the agent influence the future state of the environment. This is an important consideration in personalised suggestions (Liao et al. (2020)), which is one of the problems likely to be solved by a multitude of low-powered personal devices, fitting within our distributed processing framework. Hence we proceed to study a more sophisticated model of reinforcement learning problems in the next section, where we focus on parameter identification in full Markov Decision Processes.

## 4    Linear Value Function Prediction in Episodic MDP

We consider a Markov Decision Process (MDP) consisting of states $s \in \mathcal{S}$, actions $a \in \mathcal{A}$ and rewards the agent receives that we assume follow Assumption 3.1. We assume we observe the gameplay history of an agent taking actions according to its policy $\pi(s|a) : (\mathcal{S}, \mathcal{A}) \to [0, 1]$, hence the problem studied reduces to a Markov Reward Process. In episodic setting, we define a return of a policy $\pi$ from state $s$ as the sum of all rewards obtained while following the policy $\pi$ after having visited state $s$, i.e. $G_s = \mathbb{E}[\sum_{k=t}^{H} r_k | s_t = s]$. To evaluate how good a certain policy is in a given Markov Decision Process, it is common to learn its value function $v(s) : \mathcal{S} \to \mathbb{R}$, which assigns the return to each state. In many cases, it is sufficient to model the value function for each state as a linear combination of some features related to the given state, we will denote a vector of such features as $c_s$. We assume those features are universally known beforehand and introduce Assumption 3.2 regarding the feature vectors, similarly as in the multi-armed bandit problem.

**Assumption 4.1** *Feature vector for each state is normalised, i.e. $\|c_j\|_2^2 \leq 1$. If we form a matrix $X = (c_1, \ldots, c_n)$ with feature vectors for all states, then the smallest eigenvalue of the matrix $X^T X$ is $\eta_{min}$ such that $0 < \eta_{min} \leq 1$ and $\eta_{min} = S\lambda_{min}$ for some constant $\lambda_{min}$.*

Although in practice, a linear model will most likely be just an approximation to the true value function, within our problem setting we will assume that the problems we consider can be perfectly modelled in this way, i.e. $v(s) = c_s^T \theta$. We will now define our assumptions for episodic and non-episodic settings. We shall assume that the maximum number of transitions within an episode (so-called "horizon" length) is equal to $H$. We shall refer to the states that can be visited at step $h$ as $h$-level states and denote a set of such states as $\mathcal{S}_h$ and we denote $S_h = |\mathcal{S}_h|$. Consequently the set of initial states is $\mathcal{S}_0$ and the set of terminal states is $\mathcal{S}_H$. We introduce a different parameter vector to model the value function at each level. Hence for states at level $h$, we have that $\forall_{s \in S_h} G_s = c_s^T \theta_h$. and the parameter of interest is $\theta = \theta_h$. The inference is conducted based on the gameplay history, which consists of the steps made during $N$ episodes, i.e. $h = \{\{(s_1, a_1, r_1), \ldots, (s_{T^l}, a_{T^l}, r_{T^l})\}\}_{l=1}^{N}$, where $T^l \leq H$ is the length of the $l$th episode. Within this section we conduct our analysis under Assumption 4.2, regarding the number of times we visit each state at a given level within the history.

**Assumption 4.2** *Within gameplay history, for every state at a given level $h$ we have $E$ episodes where it was visited.*

This assumption might be treated as a strong one, however, without it, there exists a simple pathological example, where there is a special feature equal to zero in every state except for one state that is almost never visited. In such a case, no matter how many episodes we sample, unless we can guarantee that we have visited this state at least some number of times, we cannot learn the parameter value for that special feature with high certainty. We now present our lower bound on minimax risk of estimation of the parameter $\theta_h$ at level $h$ in Theorem 4.3.

**Theorem 4.3** *In a distributed offline episodic linear value function approximation problem at level $h$ and context with dimensionality $C$ and state space size of $S_h$, such that $S_h \geq C > 12$, under assumptions 3.1, 3.2 and 4.2, given $m$ processing centres, with communication budget $B \geq 1$, for any independent communication protocol, the minimax risk $M$ is lower bounded as follows:*

$$M \geq \Omega\left( \frac{C(H-h)^2 R_{max}^2}{S_h E m \lambda_{min}} \min\left\{ \max\left\{ \frac{C}{B \log m}, 1 \right\}, \frac{m}{\log m} \right\} \right)$$

**Proof 4.4** *(sketch): We present a sketch of proof here and defer the full proof to Appendix C. We proceed similarly as in the proof of Theorem 3.3 and set the $k$th component of feature vector for $j$th state to $c_{j,k} = \sqrt{\lambda_{min} S_h} \mathbb{1}_{k=j}$ for the first $S_h$ states and we set feature vectors for remaining $n - S_h$ states to zero vectors. We consider the following Markov Decision Process: starting randomly in one of the states $j$ at level $h$, the agent chooses one of two actions. Choosing the "good" action causes the agent the receive a reward of $R_{max}$ for the remaining $(H - h)$ steps until the episode ends, while the "bad" action causes the agent to receive a reward of $-R_{max}$ until the end. The agent draws a random variable $p_j \sim Bernoulli(\frac{1}{2} + \frac{c_j^T \theta}{2(H-h)R_{max}})$ and selects "good" action if $p_j = 1$ and "bad" action otherwise. We follow similar steps as in Theorem 3.3 and bound the likelihood ratio, however, under the condition that $\sum_{l=1}^{E}(2p_j^l - 1) < a$, where $a$ is a quantity we control. We then use Hoeffding inequality to bound the probability that $\sum_{l=1}^{E}(2p_j^l - 1) > a$. We then derive a bound on KL-divergence and use Lemma A.2 to derive a second bound on mutual information. We finish the proof by combining two bounds and choosing such $\delta$ and $a$ that the bound is tightest.*

Hence we get that for the risk not to depend on communication budget we need a number of at least $B > \Omega(\frac{C}{\log m})$ bits. It might be surprising that the bound decreases as the state space $S$ increases, however, because of Assumption 4.2 we have that each new state effectively increases the number of available samples for the same number of features. Note that in practice, to obtain a sensible approximation, the dimension of features is likely to grow with the size of state space.

We now show that this class of problems can be efficiently solved by using performing distributed Monte-Carlo return estimates. We present Algorithm 2, which uses Monte-Carlo to obtain return estimates and then least-squared to fit the parameters to feature vectors, which are then quantised and communicated to the main centre. We prove that it can match this lower bound with communication budget optimal up to logarithmic factors as stated by Theorem 4.5.

---

**Algorithm 2** (MC LSE) Distributed offline least-squares with Monte-Carlo return estimates

---

**Data:** $\{(s_t^l, r_t^l, a_t^l)_{t=1}^{T_l}\}_{l=1}^{N}$
On individual machines compute:
**for** $i \in \{1, \ldots, m\}$ **do**
$\quad \forall_{s \in S} \ N_s \leftarrow 0 \ G_s \leftarrow 0$
$\quad$**for** $l \in 1, \ldots, N$ **do**
$\quad\quad$**for** $s \in S$ **do**
$\quad\quad\quad f_s \leftarrow$ step of first time visit to $s$ in episode $l$
$\quad\quad\quad G_s \leftarrow G_s + \sum_{t=f_s}^{T_l} r_t^l \ N_s \leftarrow N_s + 1$
$\quad\quad$**end**
$\quad$**end**
$\quad \forall_{s \in S} \ g_s \leftarrow \frac{G_s}{N_s}$
$\quad X \leftarrow (c_1, \ldots, c_S)^T \ g \leftarrow (g_1, \ldots, g_S)^T$
$\quad \hat{\theta}^i \leftarrow (X^T X)^{-1} X^T g$
$\quad$Quantise each component of $\hat{\theta}^i$ up to precision $P$ and send to central server.
**end**
At central server compute: $\hat{\theta} \leftarrow \frac{1}{m} \sum_{i=1}^{m} \hat{\theta}^i$
**return** $\theta$

---

**Theorem 4.5** *For any value of $\theta$, Algorithm 2 using transmission with precision $P$ achieves worst-case risk upper bounded as follows:*

$$W < O\left(C \max\left\{\frac{(H-h)^2 R_{max}^2}{m S_h E \lambda_{min}}, P\right\}\right)$$

**Proof 4.6** *We observe that the return is always within the range $(-(H-h)R_{max}, (H-h)R_{max})$, hence by Popoviciu inequality we get $Var(g_s^l) \leq (H-h)^2 R_{max}^2$ and thus $Var(g_s) \leq \frac{1}{E}(H-h)^2 R_{max}^2$. Following similar steps as in the proof of Theorem 3.5 we get the statement of the Theorem.*

Hence, the number of bits transmitted $B$ must scale as $\Theta(C \log \frac{mSE\lambda_{\min}}{(H-h)R_{\max}})$ for Algorithm 2 to achieve minimax optimal performance. While the episodic setting can be used to model many problems, in applications related to electronic wearables or IoT, we would usually be interested in continuous problems, as by design those devices are meant to accompany user all the time and constantly improve their quality of life. We thus build on the results we derived so far and conduct an analysis of the non-episodic problem setting in the next section.

## 5 Linear Value Function Prediction in Non-episodic MDP

Contrary to the episodic Markov Reward Processes, in non-episodic setting we asumme all states share the same parameter vector, which defines a linear relationship between the value function and features of a state. In non-episodic setting we introduce a discount factor $\gamma \in (0,1)$, and define the return as a discounted sum of rewards starting from state $s$ and following policy $\pi$ afterwards, i.e. $G_s = \mathbb{E}[\sum_{k=0}^{\infty} r_{t+k}\gamma^k | s_t = s] = c_s^T \theta$ and $\theta$ is the parameter we would like to conduct inference about. We define the received gameplay history as i.i.d. samples of single steps made by the agent, where the initial state $s$ sampled from the stationary distribution $\pi(s)$ under its policy, i.e. $h = \{((s_t, a_t, r_t, s_{t+}))\}_{t=1}^n$, where $s_t \sim \pi(s)$. We introduce Assumption 5.1 to ensure $\pi(s)$ exists.

**Assumption 5.1** *The Markov Chain describing states visited by the agent's policy is aperiodic and irreducible, so that the stationary distribution $\pi$ exists. We assume that for each state $s \in S$ we have $\pi_{min} \le \pi(s) \le \pi_{max}$ and there is at least one state which frequently visited, i.e. $\pi_{max} > 0.01$.*

**Theorem 5.2** *In a distributed linear value function approximation problem with context dimensionality of $C > 12$, state space size of $S \le C$ and discount factor of $0 < \gamma < 0.99$, under assumptions 3.1, 4.1 and 5.1, given $m$ processing centres each with gameplay history, with each centre having a communication budget $B \ge 1$, the minimax risk $M$ of any algorithm is lower bounded as:*

$$M \ge \Omega\left(\frac{CR_{max}^2}{(1-\gamma)^2 \pi_{max} Snm\lambda_{min}} \min\left\{\max\left\{\frac{C\log m}{B}, 1\right\}, \frac{m}{\log m}\right\}\right)$$

**Proof 5.3** *(sketch): We present a sketch of proof here and defer the full proof to Appendix D.*
*We consider the following MDP: from each state $j$, regardless of the agent's action it either states in the same state with probability $1-p$ and receives a reward with a mean of $\bar{r}_j$ or moves to any other states chosen with a probability of $\frac{p}{S-1}$ and receives a reward $r_0$. We see that because of the consistency equation for value functions we have:*

$$v(j) = (1-p)\bar{r}_j + pr_0 + \gamma \sum_{s' \in S\{j\}} v(s')\frac{p}{S-1} + \gamma v(j)(1-p)$$

*We now set $r_0 = -\gamma \sum_{s' \in S\{j\}} \frac{v(s')}{S-1}$ to get that $\bar{r}_j = \frac{v(j)(1-\gamma+p\gamma)}{1-p}$. We set feature vectors and parameters in the same way as in the proof of Theorem 4.3. We also set $\bar{r}_j = (2p_j - 1)R_{max}$ where $p_j \sim Bernoulli(\frac{1}{2} + \frac{\sqrt{\lambda_{min}}Sv_k\delta}{2R_{max}}\frac{1-\gamma+p\gamma}{1-p})$. We thus have a similar situation as in the proof of Theorem 4.3, where the data received can be reduced to outcomes of Bernoulli trials. We proceed in the same way to obtain a bound that depends on $p$ and gets tighter as $p \to 0$. We now can index data generating distribution by $p$ and obtain their supremum, which gives the statement of the Theorem.*

While we can extend the lower bound to the non-episodic case while following a similar method as in the episodic one, we see that because Monte-Carlo estimators for return cannot be used in the non-episodic setting, we cannot extend Algorithm 2 in the same way. In this new setting, an algorithm needs to utilise the consistency of the value function. We thus propose Algorithm 3, which is a distributed variant of temporal difference learning. We build on the analysis conducted by Bhandari et al. (2018) to upper bound its worst-case performance in Theorem 5.4.

---

**Algorithm 3** (TD) Distributed offline temporal difference learning

---

**Data:** $(c_t, r_t, a_t, c_{t+})_{t=1}^T$
On individual machines compute:
**for** $i \in \{1, \ldots, m\}$ **do**
$\quad \hat{\theta}^i \leftarrow \theta_0$
$\quad$ **for** $t \in 1, \ldots, N$ **do**
$\quad\quad g_t \leftarrow (r_t + \gamma c_{t+}^T \hat{\theta}^i - c_t^T \hat{\theta}^i) c_t$
$\quad\quad \hat{\theta}^i \leftarrow \hat{\theta}^i + \alpha_t g_t$
$\quad$ **end**
$\quad$ Quantise each component of $\hat{\theta}^i$ up to precision $P$ and send to central server.
**end**
At central server compute: $\hat{\theta} \leftarrow \frac{1}{m} \sum_{i=1}^m \hat{\theta}^i$
**return** $\hat{\theta}$

---

**Theorem 5.4** *The worst-case risk of Algorithm 3 run with a learning rate of $\alpha_t = \frac{\beta}{\Lambda + \frac{t}{\omega}}$ with $\beta = \frac{2}{(1-\gamma)\omega}$ and $\Lambda = \frac{16}{(1-\gamma)^2 \omega}$ is upper bounded as follows:*

$$W \le O\left( \max\left\{ \frac{\max\{\frac{R_{max}^2}{S\pi_{min}\lambda_{min}m}, \|\theta - \frac{1}{m}\sum_{i=1}^m \hat{\theta}_0^i\|_2^2\}}{1 + (1-\gamma)^2 n}, CP \right\} \right)$$

**Proof 5.5** *(sketch): We present a sketch of proof here and defer the full proof to Appendix F.*
*We define $\bar{\theta}_t := \frac{1}{m} \sum_{i=1}^m \hat{\theta}^i$ to be the average vector from all machines at timestep $t$. Note that this vector is never actually created, except for the last step when we average all final results. We can see that it must satisfy the following recursive relation:*

$$\bar{\theta}_{t+1} = \frac{1}{m} \sum_{i=1}^m \hat{\theta}_{t+1}^i = \frac{1}{m} \sum_{i=1}^m [\hat{\theta}_t^i + \alpha_t g_t^i] = \bar{\theta}_t + \alpha_t \frac{1}{m} \sum_{i=1}^m g_t^i$$

*We utilise Lemmas H.1 and H.3 to show that:*

$$\mathbb{E}[\|\theta - \bar{\theta}_{t+1}\|_2^2] \le \mathbb{E}[\|\theta - \bar{\theta}_t\|_2^2](1 - 2\alpha_t \omega(1-\gamma) + \alpha_t^2) + \frac{\alpha_t^2}{m} R_{max}^2$$

*where we define $\omega$ to be the smallest eigenvalue of the covariance matrix weighted by the stationary distribution, i.e. smallest eigenvalue of $\sum_{s \in S} \pi(s) c_s c_s^T$. We then finish the proof by induction and observing that by concavity of eigenvalues we have $S\pi_{min}\lambda_{min} \le \omega \le \lambda_{min} \le 1$.*

We see that there are essentially two terms contributing to the worst-case risk, the term $\frac{R_{max}^2}{S\pi_{min}\lambda_{min}m}$ resulting from gradient's variance and the term $\|\theta - \frac{1}{m}\sum_{i=1}^m \hat{\theta}_0^i\|_2^2$ resulting from initial bias. We see that increasing the number of machines $m$ will only decrease the variance term, but will not decrease the overall worst-case risk if the initial bias is larger. This result has a practical implication, as we essentially show that using more devices within a network is not necessarily going to improve our estimate. It might spuriously appear that this algorithm is contradicting the lower bound as the worst-case risk has no explicit dependence on the dimensionality of the features. However, we can easily show (explicitly derived in Appendix G) that the initial bias will scale at least as $O(\frac{SR^2}{\lambda_{min}(1-\gamma)})$, which is consistent with the bound stated in Theorem 5.2, as we assumed $C \le S$.

In comparison to existing distributed versions of TD learning where gradients are communicated between machines at each step, in our version only the final estimates are communicated, hence the initial bias cannot be cancelled by introducing data from more machines. This is, however, due to the difficulty of our problem setting. With one communication round, it is not possible to constantly transmit information about the gradient updates. This issue would come up in practice, whenever the devices we run our algorithm on are

not constantly connected to the network and communication in real-time, during the algorithm runtime is not possible.

It might come as a surprise that although we can easily propose an optimal algorithm for the episodic case, the same is not true for the non-episodic setting. In fact, we can see that for any unbiased algorithm that has optimal risk in a non-distributed version, its distributed version averaging results from all machines will also have optimal risk as long as the lower bound scales as the inverse of the number of machines $m$. In the non-episodic setting, however, we cannot directly obtain estimates of the return and hence need to rely on a method utilising the temporal structure of the problem. Although in the non-distributed version, the temporal difference can often be superior to Monte-Carlo methods even in the episodic case due to smaller variance, we see that in a single-round communication setting, the bias is preventing the algorithm from efficiently utilising data available on all machines.

## 6 Discussions

We have studied three offline RL problems in a special case of distributed processing. We have identified a lower bound on minimax risk in each problem and proposed algorithms that match these lower bound up to logarithmic factors in the cases of contextual linear bandits and episodic MDP. In the case of non-episodic MDP, we have proposed a distributed version of temporal difference learning and analysed its worst-case risk. We have shown that its worst-case risk decreases with the number of cooperating machines only until some point, where the risk due to initial bias starts to dominate over the risk caused by variance.

Notably, our work studies a different problem from Federated Learning (Konečnỳ et al. (2015)) wherein a typical setting assumes one can send parameter values to the central server while minimising the total number of communication rounds. In our setting, we assume more restrictive conditions with regards to communication by limiting the communication round to be only one and by further restricting the number of information (i.e., the bits) each machine can send.

Moreover, we would like to emphasise that our bounds are developed on the risk in estimating the parameter values rather than on the regret (Wang et al., 2019). We note that there are problems where learning the parameter values might be of greater interest than just minimising the regret of actions. Recalling the example of physical activity suggestion in the Introduction, we not only want to learn the optimal way of suggesting the activities, but also analyse which factors influence how healthy the lifestyle of an individual is. Hence, it would be more beneficial to estimate the parameters of, for example, the value function and decrease the risk of those estimates.

In this paper, we have derived the first lower bound for distributed RL problems other bandits. Our method relies on constructing hard instances and converting them to statistical inference problems. This approach can be easily applied to other settings such as state-action value estimation or off-policy learning, which constitutes one of the directions of future work. We also identified a weakness of TD learning resulting from its initial bias. Can this bias be estimated in practice so that we do not unnecessarily utilise data from more machines than needed? Can we correct for initial bias so that worst-case risk always decreases with more machines? Can it match the derived lower bound? We leave these questions open to future research.

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

# A    General methods for deriving lower bounds

Let us define a set $\mathcal{V} = \{-1, 1\}^d$ and sample $V$ uniformly from $\mathcal{V}$. Conditioned on $V = v$ we sample $X_1$ from $P_{X_1}(\cdot | V = v)$ so that $\theta_v = \theta(P_x(\cdot | v)) = \delta v$ where $\delta > 0$ is a fixed quantity we control. Consider a Markov chain $V \to X_1 \to \cdots \to X_n \to Y \to \hat{\theta}$. Under these conditons, it was proven by Zhang et al. (2013) that for $d > 12$ we have:

$$M(\theta, P, B) \geq \delta^2 \left( \lfloor \frac{d}{6} \rfloor + 1 \right) \left( 1 - \frac{I(Y, V) + \log 2}{\frac{d}{6}} \right)$$

Hence, if we can choose a $\delta$ such that $I(Y, V)$ is upper bounded, this directly translates to a lower bound on the minimax risk. It hence remains to upper bound the mutual information any possible message $Y$ can carry about $v$, so that we can use it to obtain a lower bound on minimax risk. To that goal, we will use the inequalities described in subsequent subsections and combine them into one upper bound on mutual information. After that, it remains to find such a $\delta$ that this upper bound remains constant or is at most linear in the dimensionality of the problem.

## A.1    Tensorisation of information

**Lemma A.1 (Tensorisation of information property in Zhang et al. (2013))** *If $V$ is a parameter of distribution we wish to make inference about and the message $Y_i$ send by the ith processing centre is constructed based only on the data $X_i$ then the following is true:*

$$I(V; Y^{1:m}) \leq \sum_{i=1}^{m} I(V; Y_i)$$

## A.2    Bound on mutual information by KL-divergence

Within this section, we present another bound on mutual information, which utilises the KL-divergence of data-generating distribution.

**Lemma A.2** *Let $V \to X^i \to Y^i$ form a Markov Chain and $Y = (Y^1, \ldots, Y^m)$. Let $V$ be a $d$-dimensional vector with each component sampled uniformly from $\{v_j, v_j^*\}$. Let each $X^i$ consist of $(R_1, \ldots, R_d)$ where every $R_1 = (R_1^1, \ldots, R_1^n)$. Let every $R_j^l$ be sampled from the same distribution parametrised by $v_j$. Then:*

$$I(Y, V) \leq \frac{mn}{4} \sum_{j=1}^{d} \left( KL[p(R_j^1|v), p(R_j^1|v^*)] + KL[p(R_j^1|v^*), p(R_j^1|v)] \right)$$

**Proof A.3** *From tensorisation of information (Lemma A.1) and data-processing inequality we get:*

$$I(Y, V) \leq mI(Y^i, V) \leq mI(X^i, V)$$

*We now observe that $R_j$ are independent, hence:*

$$I((R_1, \ldots, R_d), V) = \sum_{j=1}^{d} I((R_j^1, \ldots, R_j^n), V) = \sum_{j=1}^{d} \sum_{l=1}^{n} I(R_j^l, V | R_j^{1:l-1}) =$$

$$= \sum_{j=1}^{d} \sum_{l=1}^{n} [H(R_j^l | R_j^{1:l-1}) - H(R_j^l | V, R_j^{1:l-1})] \leq \sum_{j=1}^{d} \sum_{l=1}^{n} [H(R_j^l) - H(R_j^l | V)] = n \sum_{j=1}^{d} I(R_j^1, V)$$

*Where the last inequality is true since conditioning can only reduce entropy and $R_j^i$ are independent given $V$. We know focus on the mutual information:*

$$I(R_j^1, V) = KL[p(R_j^1, V), p(R_j^1)p(V)] = KL[p(R_j^1|V)p(V), p(R_j^1)p(V)]$$

$$= \sum_v p(v)KL[p(R_j^1|v), p(R_j^1)] = \sum_v p(v)KL[\sum_{v'} p(R_j^1|v)p(v'), \sum_{v'} p(R_j^1|v')p(v')]$$

*Because of convexity of KL-divergence we get:*

$$I(R_j^1, V) \le \sum_{v,v'} p(v)p(v')KL[p(R_j^1|v), p(R_j^1|v')] =$$

$$= \frac{1}{4}\big(KL[p(R_j^1|v), p(R_j^1|v^*)] + KL[p(R_j^1|v^*), p(R_j^1|v)]\big)$$

### A.3    Bound on mutual information by set construction

We present a first bound on mutual information, which can requires constructing a set such that the samples belong to it with some probability.

**Lemma A.4 (Lemma 4 in Zhang et al. (2013))** *Let $V$ be sampled uniformly from $\{-1, 1\}^N$. For any $(i, j)$ assume that $X_j^i$ is independent of $\{V_{j'} : j' \ne j\}$ given $V_j$. Let $P_{X_j}$ be the probability measure of $X_j$ and let $B_j$ be a set such that for some $\alpha$:*

$$\sup_{S \in \sigma(B_j)} \frac{P_{X_j}(S|V = v)}{P_{X_j}(S|V = v')} \le \exp(\alpha)$$

*Define random variable $E_j = 1$ f $X_j \in B_j$ and 0 otherwise. Then*

$$I(V, Y^i) \le \sum_{j=1}^N H(E_j) + \sum_{j=1}^N P(E_j = 0) + 2(e^{4\alpha} - 1)^2 I(X^i, Y^i)$$

*Additionally if $\alpha < \frac{1.2564}{4}$, then the following is also true:*

$$I(V, Y^i) \le \sum_{j=1}^N H(E_j) + \sum_{j=1}^N P(E_j = 0) + 128\alpha^2 I(X^i, Y^i)$$

**Proof A.5** *The first statement is Lemma 4 of Zhang et al. (2013).*
*The second statement directly follows from it, as for $x < 1.2564$, it holds that $\exp(x) - 1 < 2x$*

## B    Proof of Theorem 3.3

**Theorem 3.3** *In a distributed offline linear contextual MAB problem with $A$ actions and context with dimensionality $C$ such that $CA > 12$ under assumptions 3.1 and 3.2, given $m$ processing centres each with $n \ge C$ samples for each arm, with each centre having communication budget $B \ge 1$, for any independent communication protocol, the minimax risk $M$ is lower bounded as follows:*

$$M \ge \Omega\left(\frac{ACR_{max}^2}{mn\lambda_{min}} \min\left\{\max\left\{\frac{AC}{B}, 1\right\}, m\right\}\right)$$

**Proof B.1** *Where we omit indexing by the machine index $i$, this means that statement holds for all machines. Let us consider a problem where we set the parameter vector for $j$th arm as $\boldsymbol{\theta}_j = \delta \boldsymbol{v}_j$ where $\delta$ is a quantity we will specify later, $\boldsymbol{v}_j$ is a vector sampled uniformly from $\{-1, 1\}^C$ and the $k$th element of context vector for the $l$th sample is set to $c_k^l = \sqrt{\lambda_{min}n}\mathbb{1}_{k=l}$, so that Assumption 3.2 is satisfied.*

*Let $\boldsymbol{V}$ be a concatenated vector defined as $\boldsymbol{V} = (\boldsymbol{v}_1, \ldots, \boldsymbol{v}_C)^T$ about which we would like to perform inference. Let $r_j^l$ be the reward received after pulling $j$th arm for the $l$th time. We now define the underlying process for generating the reward as follows, let $r_j^l = (2p_j^l - 1)R_{max}$, where $p_j^l = Bernoulli(\frac{1}{2} + \frac{\delta v_j^T c^l}{2R_{max}})$. We see that under such construction, the reward is always within $[-R_{max}, R_{max}]$ and its expected value is $\mathbb{E}[r_j^l] = \delta v_j^T c^l = \boldsymbol{\theta}_j c^l$.*

Let $Y^i$ be the message send by the $i$th processing centre and $X^i$ be all $r_j^l$ for all $j$ and $l$ stored on the $i$th processing centre and let us define $P^i$ in the same way for $p_j^l$. We observe that $V \to P^i \to X^i \to Y^i$ forms a Markov Chain. We see that this scenario satisfies the assumptions required to use the method from Appendix A with $d = AC$. Hence we would like to find a bound on $I(V, Y^{1:m})$. First let us consider the likelihood ratio of $p_j^l$ for $l = \{1, \ldots, n\}$ given $v_{j,k}$ and $v'_{j,k} = -v_{j,k}$. We observe that for our construction of $\mathbf{c}^k$ we essentially have $\mathbf{v}_j^T \mathbf{c}^k = c_k^k v_{j,k}$. Hence the likelihood ratio can be expressed as:

$$\left( \frac{\frac{1}{2} + \frac{c_k^k v_{j,k} \delta}{2R_{max}}}{\frac{1}{2} - \frac{c_k^k v_{j,k} \delta}{2R_{max}}} \right)^{p_j^k} \left( \frac{\frac{1}{2} - \frac{c_k^k v_{j,k} \delta}{2R_{max}}}{\frac{1}{2} + \frac{c_k^k v_{j,k} \delta}{2R_{max}}} \right)^{1 - p_j^k} = \left( \frac{\frac{1}{2} + \frac{c_k^k v_{j,k} \delta}{2R_{max}}}{\frac{1}{2} - \frac{c_k^k v_{j,k} \delta}{2R_{max}}} \right)^{2p_j^k - 1} \tag{1}$$

For $x < \frac{1}{4}$ we have that $\frac{1+x}{1-x} \le \exp\{\frac{17}{8}x\}$, hence when $\frac{c_k^k v_k \delta}{R_{max}} \le \frac{1}{4}$ (satisfied when $\frac{v_k \delta \sqrt{C\lambda_{min}}}{R_{max}} \le \frac{1}{4}$), the ratio above is bounded by:

$$\le \exp\left\{ \frac{17 c_k^k \delta v_{j,k}}{8R_{max}} \left( 2p_j^k - 1 \right) \right\} = \exp\left\{ \frac{17 \sqrt{n\lambda_{min}} \delta v_k}{8R_{max}} \left( 2p_j^k - 1 \right) \right\}$$

We observe that we always have $\left| 2p_j^k - 1 \right| \le 1$, hence the ratio is bounded by $\exp\{\frac{17\delta\sqrt{n\lambda_{min}}}{8R_{max}}\}$. We can thus satisfy the conditions of Lemma A.4 with $\alpha = \frac{17\delta\sqrt{n\lambda_{min}}}{8R_{max}}$ if we define $B_{j,k} = \{p_j^k : 2p_j^k - 1 \le 1\}$, where we have that $P(p_j^k \in B_{j,k}) = 1$. If we can set $\delta$ such that $\frac{17\delta\sqrt{n\lambda_{min}}}{8R_{max}} < \frac{1.2564}{4}$, then we get:

$$I(V, Y^i) \le \frac{289\lambda_{min}n\delta^2}{64R_{max}^2} I(X^i, Y^i)$$

Note that we use a two-dimensional index for $X$, whereas in Lemma A.4, the index is one dimensional. This is just a matter of notation and those two types of indexing are mathematically equivalent. We see that we can bound $I(X^i, Y)$ as follows:

$$I(X^i, Y^i) \le H(Y^i) \le B$$

Combining this with the tensorisation of information (Lemma A.1) we get:

$$I(V, Y) \le \frac{289\lambda_{min}mn\delta^2}{64R_{max}^2} B \tag{2}$$

We now develop a second inequality for $I(V, Y)$. We observe for the distribution of $p_k^k$ we have the following:

$$KL[p(p_k^k|v), p(p_k^k|v^*)] = \left( \frac{1}{2} + \frac{c_k^k \delta}{2R_{max}} \right) \log\left( \frac{\frac{1}{2} + \frac{c_k^k \delta}{2R_{max}}}{\frac{1}{2} - \frac{c_k^k \delta}{2R_{max}}} \right) + \left( \frac{1}{2} - \frac{c_k^k \delta}{2R_{max}} \right) \log\left( \frac{\frac{1}{2} - \frac{c_k^k \delta}{2R_{max}}}{\frac{1}{2} + \frac{c_k^k \delta}{2R_{max}}} \right) =$$

$$= \left( \frac{1}{2} + \frac{c_k^k \delta}{2R_{max}} - \frac{1}{2} + \frac{c_k^k \delta}{2R_{max}} \right) \log\left( \frac{\frac{1}{2} + \frac{c_k^k \delta}{2R_{max}}}{\frac{1}{2} - \frac{c_k^k \delta}{2R_{max}}} \right) = \frac{c_k^k \delta}{R_{max}} \log\left( \frac{\frac{1}{2} + \frac{c_k^k \delta}{2R_{max}}}{\frac{1}{2} - \frac{c_k^k \delta}{2R_{max}}} \right)$$

Same as before we use the fact that for $x < \frac{1}{4}$ we have that $\frac{1+x}{1-x} \le \exp\{\frac{17}{8}x\}$, we get:

$$KL[p(p_k^k|v), p(p_k^k|v^*)] \le \frac{17}{8} \frac{(c_k^k)^2 \delta^2}{R_{max}^2} = \frac{17}{8} \frac{n\lambda_{min}\delta^2}{R_{max}^2} \tag{3}$$

Hence by Lemma A.2 with $d = AC$ and identifying $R_k = (p_k^k)$ we get:

$$I(V, Y) \le \frac{17}{8} \frac{mnAC\delta^2\lambda_{min}}{R_{max}^2} \tag{4}$$

*Combining inequalities 2 and 4 we get:*

$$I(V, Y) \leq \frac{17}{8} \frac{\delta^2 mn\lambda_{min}}{R_{max}^2} \min \left\{ \frac{17}{8} B, \frac{CA}{2} \right\}$$

*We can now set:*

$$\delta_A^2 \leq \frac{1}{10} \frac{CAR_{max}^2}{\frac{17}{8} mn\lambda_{min} \min \left\{ \frac{17}{8} B, \frac{CA}{2} \right\}} = \frac{1}{10} \frac{R_{max}^2}{\frac{17}{8} mn\lambda_{min} \min \left\{ \frac{17}{8} \frac{B}{CA}, \frac{1}{2} \right\}}$$

$$\delta_B^2 \leq \frac{1}{10} \frac{R_{max}^2}{\frac{17}{8} n\lambda_{min}}$$

$$\delta = \min\{\delta_A, \delta_B\}$$

*We see that under such construction we have $\frac{\delta\sqrt{n\lambda_{min}}}{R_{max}} < \frac{1}{4}$ and $\frac{17\delta\sqrt{n\lambda_{min}}}{8R_{max}} < \frac{1.2564}{4}$ and we can also bound the mutual information as follows:*

$$I(V, Y) \leq \frac{CA}{10}$$

*We now remind the Assumption of the Theorem that $AC > 12$, which allows us to show:*

$$\left(1 - \frac{I(Y, V) + \log 2}{\frac{AC}{6}}\right) \geq \left(0.65 - \frac{I(Y, V)}{\frac{AC}{6}}\right) \geq \left(0.65 - \frac{6}{10}\right) \geq 0 \tag{5}$$

*Which together with the method from Appendix A completes the proof.*

## C    Proof of Theorem 4.3

**Theorem 4.3** *In a distributed offline episodic linear value function approximation problem at level $h$ and context with dimensionality $C$ and state space size of $S_h$, such that $S_h \geq C > 12$, under assumptions 3.1, 3.2 and 4.2, given $m$ processing centres, with communication budget $B \geq 1$, for any independent communication protocol, the minimax risk $M$ is lower bounded as follows:*

$$M \geq \Omega\left(\frac{C(H-h)^2 R_{max}^2}{S_h Em\lambda_{min}} \min\left\{\max\left\{\frac{C}{B\log m}, 1\right\}, \frac{m}{\log m}\right\}\right)$$

**Proof C.1** *We consider the following Markov Decision Process: starting randomly in one of the states $i$ at level $h$, the agent chooses one of two actions. Choosing the "good" action causes the agent the receive a reward of $R_{max}$ for the remaining $(H - h)$ steps until the episode ends. Choosing the "bad" action causes the agent to receive a reward of $-R_{max}$ until the end of episode. The agent draws a random variable $p_j \sim$ Bernoulli$(\frac{1}{2} + \frac{c_j^T \theta}{2(H-h)R_{max}})$ and selects "good" action if $p_j = 1$ and "bad" action otherwise.*

*We see that with such construction, the maximum absolute value of the reward can never be greater than $R_{max}$ and the mean of the return from state $j$ is $\mathbb{E}[G_j] = \mathbb{E}[\sum_{t=h}^{H} r_j(t)] = \mathbb{E}[\sum_{t=h}^{H} r_j(t)|p_j = 1]P(p_j = 1) + \mathbb{E}[\sum_{t=h}^{H} r_j(t)|p_j = 0]P(p_j = 0) = c_j^T \theta$ and hence the value function of state $j$ is $v(j) = c_j^T \theta$.*

*Let us now define the parameters as $\theta = \delta v$, where $v$ is sampled uniformly from $\{-1, 1\}^C$. For every episode, gameplay history contains $(H - h)$ tuples $(c_t, a_t, r_t)$. Hence, we can just treat the received information as $E$ samples of $p_j$ and context $c_j$ for each state $j \in S_h$. We can see that this construction satisfies the problem setting of Appendix A, with $d = C$. We select context elements as $c_{j,k} = \sqrt{S\lambda_{min}}\mathbb{1}_{j=k}$ so that Assumption 3.2 is satisfied. Let us now consider the likelihood ratio of $\{p_k^l\}_{l=1}^E$ given $v_k = -v_k'$, defining $n_k^l = 1 - p_k^l$ we have:*

$$\prod_{l=1}^E \left(\frac{\frac{1}{2} + \frac{c_{j,k}v_k\delta}{2R_{max}(H-h)}}{\frac{1}{2} - \frac{c_{j,k}v_k\delta}{2R_{max}(H-h)}}\right)^{p_k^l} \left(\frac{\frac{1}{2} - \frac{c_{j,k}v_k\delta}{2R_{max}(H-h)}}{\frac{1}{2} + \frac{c_{j,k}v_k\delta}{2R_{max}(H-h)}}\right)^{1-p_k^l} = \prod_{l=1}^E \left(\frac{\frac{1}{2} + \frac{c_{j,k}v_k\delta}{2R_{max}(H-h)}}{\frac{1}{2} - \frac{c_{j,k}v_k\delta}{2R_{max}(H-h)}}\right)^{p_k^l - n_k^l}$$

For $x < \frac{1}{4}$ we have that $\frac{1+x}{1-x} \leq \exp\{\frac{17}{8}x\}$, hence when $\frac{c_{j,k} v_k \delta}{R_{max}(H-h)} \leq \frac{1}{4}$ (satisfied when $\frac{v_k \delta \sqrt{S \lambda_{min}}}{R_{max}(H-h)} \leq \frac{1}{4}$), the ratio above is bounded by:

$$\leq \exp\left\{\frac{17\delta v_k}{8 R_{max}(H-h)} \sum_{l=1}^{E} c_{j,k}\left[p_k^l - n_k^l\right]\right\} \leq \exp\left\{\frac{17\sqrt{S\lambda_{min}}\delta v_k}{8 R_{max}(H-h)} \left|\sum_{l=1}^{E} p_k^l - n_k^l\right|\right\} \tag{6}$$

If it holds that $\left|\sum_{l=1}^{E} p_k^l - n_k^l\right| \leq a$ then we have that the ratio is bounded by $\exp\{\frac{17\delta\sqrt{\lambda_{min}}}{8 R_{max}(H-h)} a\}$. Hence we can satisfy the conditions of Lemma A.4 with $\alpha = \frac{17\delta\sqrt{S\lambda_{min}}a}{8 R_{max}(H-h)}$ if we define $B_k$ as:

$$B_k = \left\{(p_k^l, \ldots, p_k^l)\}_{l=1}^{E} \in \mathbb{Z}_+^E : \left|\sum_{l=1}^{E} p_k^l - n_k^l\right| < a\right\}$$

To complete proof it remains to bound:

$$P(E_k = 0) = \frac{1}{2}\left(P\left(\sum_{l=1}^{E} p_k^l - n_k^l > a \middle| v_k = 1\right) + P\left(\sum_{l=1}^{E} p_k^l - n_k^l > a \middle| v_k = -1\right)\right.$$

$$\left. + P\left(\sum_{l=1}^{E} p_k^l - n_k^l < -a | v_k = 1\right) + P\left(\sum_{l=1}^{E} p_k^l - n_k^l < -a \middle| v_k = -1\right)\right)$$

We now notice that due to symmetry, the first and fourth time are equal and so are second and third. We also notice that the first term must be greater or equal to the second. This gives:

$$P(E_k = 0) \leq 2P\left(\sum_{l=1}^{E} p_k^l - n_k^l > a \middle| v_k = 1\right)$$

We notice that the mean of $\sum_{l=1}^{E} p_k^l - n_k^l$ is $\mu_k = \frac{v_k \delta}{(H-h)R_{max}} \sum_{l=1}^{E} c_k = \frac{\delta E v_k \sqrt{S\lambda_{min}}}{(H-h)R_{max}}$. We can subtract it from both side of inequality and divide them by $E$:

$$P(E_k = 0) \leq 2P\left(\frac{1}{E}\left[\sum_{l=1}^{E}\sum_{t=H-h}^{H} p_k^l(t) - n_k^l(t)\right] - \frac{\mu_k}{E} > \frac{a}{E} - \frac{\mu_k}{E} \middle| v_k = 1\right)$$

Since $p_k^l(t)$ are independent given $v_k$, we can use Hoeffding inequality with the number of variables equal to $E$, each being confined to $[0, 1]$. Thus if $a > \mu_k$:

$$P(E_k = 0) \leq 2\exp\left\{-\frac{2E^2(\frac{a}{E} - \frac{\mu_k}{E})^2}{E}\right\} = 2\exp\left\{-\frac{2(a - \mu_k)^2}{E}\right\}$$

$$= 2\exp\left\{-\frac{2\left(a - \delta\frac{E\sqrt{S\lambda_{min}}}{(H-h)R_{max}}\right)^2}{E}\right\} \tag{7}$$

Same as in Equation 3, we can bound the KL-divergence as follows:

$$KL[p(p_k^l|v), p(p_k^l|v^*)] \leq \frac{17}{8}\frac{(c_k^k)^2\delta^2}{(H-h)^2 R_{max}^2} = \frac{17}{8}\frac{\lambda_{min}\delta^2}{(H-h)^2 R_{max}^2}$$

We now use Lemma A.2 with $d = C$ and identifying $R_k^l = p_k^l$ to obtain:

$$I(Y, V) \leq \frac{17 m S E}{16}\frac{C\lambda_{min}}{(H-h)^2 R_{max}^2}\delta^2 \tag{8}$$

*Combining inequalities 6, 7 and 8 we get:*

$$I(Y, V) \leq \frac{17\delta^2 m\lambda_{min}}{8(H-h)^2 R_{max}^2} \min\left\{\frac{17}{8}a^2 B, \frac{SEC}{2}\right\} +$$

$$+Cmh\left(2\exp\left\{-\frac{2\left(a - \delta\frac{E\sqrt{S\lambda_{min}}}{2(H-h)R_{max}}\right)^2}{E}\right\}\right) + 2Cm\exp\left\{-\frac{2\left(a - \delta\frac{E\sqrt{S\lambda_{min}}}{2(H-h)R_{max}}\right)^2}{E}\right\}$$

*We can now set:*

$$\delta_A^2 \leq \frac{1}{100}\frac{8C(H-h)^2 R_{max}^2}{17m\lambda_{min}SE\min\{\frac{17}{4}\frac{Ba^2}{E}, \frac{C}{2}\}} = \frac{1}{100}\frac{8C(H-h)^2 R_{max}^2}{17m\lambda_{min}SE\min\{\frac{17}{4}\frac{Ba^2}{CE}, \frac{1}{2}\}}$$

$$\delta_B^2 \leq \frac{8R_{max}^2(H-h)^2}{17ES\lambda_{min}\log m}$$

$$\delta = \min\{\delta_A, \delta_B\}$$

$$a = 100\sqrt{E\log 100m}$$

*We see that under such choice we have* $\frac{v_k\delta\sqrt{S\lambda_{min}}}{R_{max}(H-h)} \leq \frac{1}{4}$ *and* $a > \mu_k$. *We now have that:*

$$\frac{17\delta_A^2 m\lambda_{min}}{8(H-h)^2 R_{max}^2}\min\left\{\frac{17}{8}a^2 B, \frac{SEC}{2}\right\} \leq 0.01C$$

*Since* $a > \mu_k$ *we have that:*

$$(a - \delta_B\frac{E\sqrt{S\lambda_{min}}}{2(H-h)R_{max}})^2 \geq (100\sqrt{E\log 100m} - \frac{\sqrt{E}}{\log m})^2$$

$$= 100E(1 - \frac{1}{\log m})^2\log 100m \geq E\log 100m$$

*We can use this to bound the third term as follows:*

$$2mC\exp\left\{-\frac{2\left(a - \delta_B\frac{\sqrt{S\lambda_{min}}}{2(H-h)R_{max}}\right)^2}{E}\right\} \leq 2mC\exp\{-\frac{2E\log 100m}{E}\} = 2mC\exp\{-2\log 100m\} \leq \frac{0.0002C}{m}$$

*For the second term we proceed similarly by observing that for binary entropy function we have* $h(q) \leq 6/5\sqrt{q}$ *for* $q > 0$, *which gives:*

$$mCh\left(2\exp\left\{-\frac{2\left(a - \delta_B\frac{\sqrt{S\lambda_{min}}}{2(H-h)R_{max}}\right)^2}{E}\right\}\right) \leq mCh(\frac{0.0002}{m^2}) \leq \frac{6}{5}C\sqrt{0.0002} \leq 0.02C$$

*Hence we get that:*

$$I(Y, V) \leq 0.01C + \frac{0.0002C}{m} + 0.02C \leq 0.04C \leq 0.10C$$

*We can now obtain an inequality similar to the one in Equation 5 and proceed as in the proof of Theorem 3.3 mutatis mutandis, which completes the proof.*

## D  Proof of Theorem 5.2

**Theorem 5.2** *In a distributed linear value function approximation problem with context dimensionality of $C > 12$, state space size of $S \leq C$ and discount factor of $0 < \gamma < 0.99$, under assumptions 3.1, 4.1 and 5.1, given $m$ processing centres each with gameplay history, with each centre having a communication budget $B \geq 1$, the minimax risk $M$ of any algorithm is lower bounded as:*

$$M \geq \Omega\left(\frac{CR_{max}^2}{(1-\gamma)^2 \pi_{max} Snm\lambda_{min}} \min\left\{\max\left\{\frac{C\log m}{B}, 1\right\}, \frac{m}{\log m}\right\}\right)$$

**Proof D.1** *We consider the following MDP: from each state $i$, regardless of the agent's action it either states in the same state with probability $1 - p$ and receives a reward $r_j$ with a mean of $\bar{r}_j$ or moves to any other states chosen with a probability of $\frac{p}{S-1}$ and receives a reward $r_0$. We see that because of the consistency equation for value functions we have:*

$$v(i) = (1-p)\bar{r}_i + pr_0 + \gamma \sum_{s' \in S\{i\}} v(s')\frac{p}{S-1} + \gamma v(i)(1-p)$$

*Similarily as in previous proofs, we set the $k$-th element of context vector for the $i$th state as $c_{i,k} = \sqrt{\lambda_{min}S}\mathbb{1}_{k=i}$ so that Assumption 3.2 is satisfied. We also set the $k$th element of parameter vector to be $\theta_k = \delta v_k$, where $v_k$ is sampled uniformly from $\{-1,1\}$. We now set $r_0 = -\gamma \sum_{s' \in S\{i\}} \frac{v(s')}{S-1}$ to get that:*

$$v(i) = (1-p)\bar{r}_i + \gamma v(i)(1-p)$$

$$\bar{r}_i = \frac{v(i)(1-\gamma+p\gamma)}{1-p}$$

*For states $i > C$ we can just deterministicaly set $r_i = 0$, for other states we get that:*

$$\bar{r}_i = \frac{\sqrt{\lambda_{min}S}(1-\gamma+p\gamma)}{1-p}$$

*We now set $r_i = \frac{1}{99}(2p_i - 1)R_{max}$ where $p_i \sim Bernoulli(\frac{1}{2} + \frac{99\sqrt{\lambda_{min}S}v_k\delta}{2R_{max}}\frac{1-\gamma+p\gamma}{1-p})$. We see that under this construction expected return from each state is $c^T\theta$ and $|r_i| \leq R_{max}$. Since under this construction the maximum value of the value function for any state $s$ is $v(s) \leq \frac{0.99R_{max}}{1-\gamma}$, we also have that $|r_0| \leq \gamma\frac{0.99R_{max}}{1-\gamma} \leq R_{max}$. We thus have a similar situation as in the proof of Theorem 4.3, where the data received can be reduced to outcomes of Bernoulli trials. Same as in that proof, we can show that the likelihood ratio given $v_k = -v'_k$ is bounded by $\exp\left\{\frac{17\sqrt{S\lambda_{min}}\delta v_k}{8R_{max}}\frac{1-\gamma+p\gamma}{1-p}\left|\sum_{l=1}^{N_k} p_k^l - n_k^l\right|\right\}$, where $N_k$ is the number of visits to state $k$ in gameplay history. We notice that the mean of $\sum_{l=1}^{N_k} p_k^l - n_k^l$ is $\mu_k = \frac{99\delta\pi(k)nv_k\sqrt{S\lambda_{min}}}{R_{max}}\frac{1-\gamma+p\gamma}{1-p}$ and we can derive a similar Hoeffding bound as in Equation 7 with $n$ variables confined to $[0,1]$, where if $N_k < n$ for the last $N_k - n$ variables we set $p_k^l = n_k^l = 0$. This gives us the following inequality:*

$$P(E_k = 0) \leq 2\exp\left\{-\frac{2\left(a - \delta\frac{99n\pi(k)\sqrt{S\lambda_{min}}}{R_{max}}\frac{1-\gamma+p\gamma}{1-p}\right)^2}{n}\right\}$$

*We can also develop a bound on KL divergence. We denote by $P_k$ all $p_k^l$ for $l = 1, \ldots, N_k$ and note that through convexity of KL divergence we get:*

$$KL[p(P_k|v), p(P_k|v*)] \leq \sum_{N_k=1}^{n} KL[p(P_k|v, N_k), p(P_k|v*, N_k)]p(N_k)$$

*Since distribution of $P_k$ is Binomial($n, \pi(k)$) we get:*

$$KL[p(P_k|v), p(P_k|v*)] \leq \sum_{N_k=1}^{n} p(N_k)\log\left(\frac{\frac{1}{2}+q}{\frac{1}{2}-q}\right)N_k q$$

Where we used $q = \frac{99\sqrt{\lambda_{min}}Sv_k\delta}{2R_{max}}\frac{1-\gamma+p\gamma}{1-p}$. Same as before we use the fact that for $x < \frac{1}{4}$ we have that $\frac{1+x}{1-x} \leq \exp\{\frac{17}{8}x\}$:

$$KL[p(P_k|v), p(P_k|v*)] \leq \frac{17}{8}q^2 \sum_{N_k=1}^{n} p(N_k)N_k = \frac{17}{8}q^2 n\pi(k) \leq \frac{17}{8}q^2 n\pi_{max}$$

From Lemma A.2 with $d = C$ and $R_k = P_k$ we get:

$$I(Y,V) \leq \frac{Cm}{2}\left(\frac{17}{8}\frac{99\sqrt{\lambda_{min}}Sv_k\delta}{2R_{max}}\frac{1-\gamma+p\gamma}{1-p}\right)^2 n\pi_{max}$$

Proceeding as in the proof of Theorem 4.3 mutandits mutatis we get that:

$$I(Y,V) \leq \frac{17\delta^2 mS\lambda_{min}}{8R_{max}^2}\left(\frac{1-\gamma+p\gamma}{1-p}\right)^2 \min\left\{\frac{17}{8}a^2 B, \frac{Cn\pi_{max}}{2}\right\}+$$

$$+Cmh\left(2\exp\left\{-\frac{2\left(a-\delta\frac{n\pi(k)\sqrt{S\lambda_{min}}}{R_{max}}\frac{1-\gamma+p\gamma}{1-p}\right)^2}{n}\right\}\right) + Cm2\exp\left\{-\frac{2\left(a-\delta\frac{n\pi(k)\sqrt{S\lambda_{min}}}{R_{max}}\frac{1-\gamma+p\gamma}{1-p}\right)^2}{n}\right\}$$

Hence we can set:

$$a \propto \sqrt{n\pi_{max}\log m}$$

$$\delta_a^2 \propto \frac{CR_{max}^2}{m\lambda_{min}S\min\{Ba^2, Cn\pi_{max}\}}\left(\frac{1-\gamma+p\gamma}{1-p}\right)^2$$

$$\delta_b^2 \propto \frac{R_{max}^2}{nS\pi_{max}\lambda_{min}\log m}\left(\frac{1-\gamma+p\gamma}{1-p}\right)^2$$

$$\delta = \min\{\delta_a, \delta_b\}$$

Proceeding similarly as in the proof of Theorem 4.3 we get:

$$M \geq \Omega\left(\frac{CR_{max}^2(1-p)^2}{(1-\gamma+p\gamma)^2 S\pi_{max}nm\lambda_{min}}\min\left\{\max\left\{\frac{C\log m}{B}, 1\right\}, \frac{m}{\log m}\right\}\right)$$

We observe that the bound gets tighter as $p \to 0$. We can now index distributions for $r_i$ with $p$ and observe that since the bound is defined as a supremum over the set of considered distributions, we get the statement of the Theorem.

## E    Proof of Theorem 3.5

**Theorem 3.5** *Let us define $\boldsymbol{\theta} = (\boldsymbol{\theta}_1, \ldots, \boldsymbol{\theta}_A)$ to be the concatenated parameter vector for all actions. For any value of $\boldsymbol{\theta}$, Algorithm 1 using transmission with precision $P$ achieves a worst-case risk upper bounded as follows:*

$$W < O\left(AC\max\left\{\frac{R_{max}^2}{mn\lambda_{min}}, P\right\}\right)$$

**Proof E.1** *We start by assuming the transmission is lossless (i.e. the number of bits is infinite) and then study how the MSE changes when transmission introduces quantisation error. Using the bias variance decomposition, we get:*

$$MSE(\hat{\boldsymbol{\theta}}) = \sum_{a\in A}\sum_{k=1}^{C} bias(\hat{\theta}_{a,k})^2 + \sum_{a\in A}\sum_{k=1}^{C} Var(\hat{\theta}_{a,k})$$

*It is a well-known fact that for least-squares estimate, the bias is zero, hence $bias(\hat{\theta}_{a,k}^i) = 0$ and their average $\hat{\theta}_{a,k}$ is also unbiased. Because of the properties of the variance of average values we get:*

$$MSE(\hat{\boldsymbol{\theta}}) = \frac{1}{m}\sum_{a\in A}\sum_{k=1}^{C} Var(\hat{\theta}_{a,k}^1)$$

*Using eigendecomposition and singular value decomposition we get:*

$$X_a^T X_a = Q^T \Lambda Q$$

$$X_a^T = Q^T \Sigma V^T$$

*Where we choose such $Q$ and $V$ such that all left and right singular vectors have a norm of 1. Since $Q$ is orthogonal, it follows that:*

$$\hat{\boldsymbol{\theta}}_a^i = (Q^T \Lambda Q)^{-1} Q^T \Sigma V^T \boldsymbol{r}_a = Q^T \Lambda^{-1} Q Q^T \Sigma V^T \boldsymbol{r}_a = Q^T \Lambda^{-1} \Sigma V^T \boldsymbol{r}_a$$

*Writing this equation in terms of matrix elements gives:*

$$\hat{\theta}_{a,k}^i = \sum_{j=1}^{C} Q_{j,k} \frac{\sqrt{\lambda_j}}{\lambda_j} \sum_{l=1}^{n} r_a^l V_{j,l} = \sum_{j=1}^{C} \sum_{l=1}^{n} Q_{j,k} \frac{1}{\sqrt{\lambda_j}} r_a^l V_{j,l} \leq \sum_{j=1}^{C} \sum_{l=1}^{n} Q_{j,k} \frac{1}{\sqrt{\lambda_{min}n}} r_a^l V_{j,l}$$

*Where $\lambda_i$ are the eigenvalues. The last inequality follows from Assumption 3.2. We can now easily bound the variance:*

$$Var(\hat{\theta}_{a,k}^i) \leq \sum_{j=1}^{C} \sum_{l=1}^{n} \frac{Q_{j,k}^2 V_{j,l}^2}{\lambda_{min}n} Var(r_a^l)$$

*We now observe that due to Popoviciu inequality and Assumption 3.1 we get: $Var(r_a^l) \leq R_{max}^2$. This gives us:*

$$Var(\hat{\theta}_{a,k}^i) \leq \frac{R_{max}^2}{\lambda_{min}n} \sum_{j=1}^{C} Q_{j,k}^2 \sum_{l=1}^{n} V_{j,l}^2 = \frac{R_{max}^2}{\lambda_{min}n}$$

*Where the last equality is due to singular vectors having norm of 1. Substituting this back into the equation for MSE, we get:*

$$MSE(\hat{\boldsymbol{\theta}}) \leq \frac{AC R_{max}^2}{mn\lambda_{min}}$$

*Using the fact that each component is quantised up to precision $P$, the max error introduced by quantisation is $ACP$, hence:*

$$MSE(\hat{\boldsymbol{\theta}}) \leq O\left(AC \max\left\{\frac{R_{max}^2}{mn\lambda_{min}}, P\right\}\right)$$

## F Proof of Theorem 5.4

**Theorem 5.4** *The worst-case risk of Algorithm 3 run with a learning rate of $\alpha_t = \frac{\beta}{\Lambda + \frac{t}{\omega}}$ with $\beta = \frac{2}{(1-\gamma)\omega}$ and $\Lambda = \frac{16}{(1-\gamma)^2 \omega}$ is upper bounded as follows:*

$$W \leq O\left(\max\left\{\frac{\max\{\frac{R_{max}^2}{S\pi_{min}\lambda_{min}m}, \|\boldsymbol{\theta} - \frac{1}{m}\sum_{i=1}^{m} \hat{\boldsymbol{\theta}}_0^i\|_2^2\}}{1 + (1-\gamma)^2 n}, CP\right\}\right)$$

**Proof F.1** *We define $\bar{\boldsymbol{\theta}}_t := \frac{1}{m} \sum_{i=1}^{m} \hat{\boldsymbol{\theta}}_t^i$ to be the average vector from all machines at timestep $t$. Note that this vector is never actually created, except for the last step when we average all final results. We define $\omega$ to be the smallest eigenvalue of the covariance matrix weighted by the stationary distribution, i.e. smallest eigenvalue of $\sum_{s \in S} \pi(s) \boldsymbol{c}_s \boldsymbol{c}_s^T$. By convexity of eigenvalues we have $S\pi_{min}\lambda_{min} \leq \omega \leq \lambda_{min} \leq 1$. We can see that the averaged parameter vector must satisfy the following recursive relation:*

$$\bar{\boldsymbol{\theta}}_{t+1} = \frac{1}{m} \sum_{i=1}^{m} \hat{\boldsymbol{\theta}}_{t+1}^i = \frac{1}{m} \sum_{i=1}^{m} [\hat{\boldsymbol{\theta}}_t^i + \alpha_t \boldsymbol{g}_t^i] = \bar{\boldsymbol{\theta}}_t + \alpha_t \frac{1}{m} \sum_{i=1}^{m} \boldsymbol{g}_t^i$$

*Hence we can write:*

$$\mathbb{E}[\|\boldsymbol{\theta} - \bar{\boldsymbol{\theta}}_{t+1}\|_2^2] = \mathbb{E}[\|\boldsymbol{\theta} - \bar{\boldsymbol{\theta}}_t\|_2^2] - \frac{2\alpha_t}{m} \mathbb{E}\left[\sum_{i=1}^{m} (\boldsymbol{\theta} - \bar{\boldsymbol{\theta}}_t^i)^T \boldsymbol{g}_t^i\right] + \alpha_t^2 \mathbb{E}\left[\frac{1}{m^2} \|\sum_{i=1}^{m} \boldsymbol{g}_t^i\|_2^2\right]$$

We now focus on the second term:

$$\frac{2\alpha_t}{m} \sum_{i=1}^{m} \mathbb{E}\Big[(\boldsymbol{\theta} - \bar{\boldsymbol{\theta}}_t^i)^T \boldsymbol{g}_t^i\Big] = \frac{2\alpha_t}{m^2} \sum_{i=1}^{m} \sum_{j=1}^{m} \mathbb{E}\Big[(\boldsymbol{\theta} - \hat{\boldsymbol{\theta}}_t^j)^T \mathbb{E}\big[\boldsymbol{g}_t^i \big| \hat{\boldsymbol{\theta}}_t^i\big]\Big]$$

using Lemma H.1 we have that

$$\frac{2\alpha_t}{m^2} \sum_{i=1}^{m} \sum_{j=1}^{m} \mathbb{E}\Big[(\boldsymbol{\theta} - \hat{\boldsymbol{\theta}}_t^j)^T \mathbb{E}\big[\boldsymbol{g}_t^i \big| \hat{\boldsymbol{\theta}}_t^i\big]\Big] \geq \frac{2\alpha_t \omega(1 - \gamma)}{m^2} \mathbb{E}\Big[ \sum_{i=1}^{m} \sum_{j=1}^{m} \|\boldsymbol{\theta} - \boldsymbol{\theta}_t^i\|_2 \|\boldsymbol{\theta} - \boldsymbol{\theta}_t^j\|_2\Big]$$

We now decompose the third term, while using the notation $\hat{\boldsymbol{\theta}}_t = (\hat{\boldsymbol{\theta}}_t^1, \ldots, \hat{\boldsymbol{\theta}}_t^m)$:

$$\mathbb{E}[\|\frac{1}{m} \sum_{i=1}^{m} \boldsymbol{g}_t^i\|_2^2] = \mathbb{E}[\mathbb{E}[\|\frac{1}{m} \sum_{i=1}^{m} \boldsymbol{g}_t^i\|_2^2 | \hat{\boldsymbol{\theta}}_t, \boldsymbol{c}_t^i, \boldsymbol{c}_{t+}^i]]$$

$$= \mathbb{E}[\frac{1}{m^2} \sum_{i=1}^{m} \sum_{j=1}^{m} \mathbb{E}[\boldsymbol{g}_t^i | \hat{\boldsymbol{\theta}}_t, \boldsymbol{c}_t^i, \boldsymbol{c}_{t+}^i]^T \mathbb{E}[\boldsymbol{g}_t^j | \hat{\boldsymbol{\theta}}_t, \boldsymbol{c}_t^i, \boldsymbol{c}_{t+}^i] + Tr(\frac{1}{m^2} \sum_{i=1}^{m} Var(\boldsymbol{g}_t^i | \hat{\boldsymbol{\theta}}_t, \boldsymbol{c}_t^i, \boldsymbol{c}_{t+}^i))]$$

The expected value can be upper-bounded as follows:

$$\mathbb{E}[\boldsymbol{g}_t^i] = \mathbb{E}[(r_t^i + \gamma (\boldsymbol{c}_{t+}^i)^T \boldsymbol{\theta}_t^i - (\boldsymbol{c}_t^i)^T \boldsymbol{\theta}_t^i) \boldsymbol{c}_t^i] =$$

$$= \mathbb{E}[(r_t^i + \gamma (\boldsymbol{c}_{t+}^i)^T \boldsymbol{\theta} - \boldsymbol{c}_t^i \boldsymbol{\theta} + \gamma (\boldsymbol{c}_{t+}^i)^T (\boldsymbol{\theta}_t^i - \boldsymbol{\theta}) - (\boldsymbol{c}_t^i)^T (\boldsymbol{\theta}_t^i - \boldsymbol{\theta})) \boldsymbol{c}_t^i] =$$

$$= (\mathbb{E}[r_t^i] + \gamma \boldsymbol{\theta}^T \boldsymbol{c}_{t+}^i - \boldsymbol{\theta}^T \boldsymbol{c}_t^i) \boldsymbol{c}_t^i + (\gamma \boldsymbol{c}_{t+}^i - \boldsymbol{c}_t^i)^T (\boldsymbol{\theta}_t^i - \boldsymbol{\theta}) \boldsymbol{c}_t^i$$

By the definition of value function we have that $\mathbb{E}[r_t^i] = \boldsymbol{\theta}^T \boldsymbol{c}_t^i - \gamma \boldsymbol{\theta}^T \boldsymbol{c}_{t+}^i$. Hence the expression above simplifies to $(\gamma \boldsymbol{c}_{t+}^i - \boldsymbol{c}_t^i)^T (\boldsymbol{\theta}_t^i - \boldsymbol{\theta}) \boldsymbol{c}_t^i$. We thus have:

$$\mathbb{E}[g_t^i]^T \mathbb{E}[g_t^j] = (\gamma \boldsymbol{c}_{t+}^i - \boldsymbol{c}_t^i)^T (\boldsymbol{\theta}_t^i - \boldsymbol{\theta}) (\boldsymbol{c}_t^i)^T \boldsymbol{c}_t^j (\gamma \boldsymbol{c}_{t+}^j - \boldsymbol{c}_t^j)^T (\boldsymbol{\theta}_t^j - \boldsymbol{\theta}) \leq$$

$$\leq \|\gamma \boldsymbol{c}_{t+}^i - \boldsymbol{c}_t^i\|_2 \|\boldsymbol{\theta}_t^i - \boldsymbol{\theta}\|_2 \|\boldsymbol{c}_t^i\|_2 \|\boldsymbol{c}_t^j\|_2 \|\gamma \boldsymbol{c}_{t+}^j - \boldsymbol{c}_t^j\|_2 \|\boldsymbol{\theta}_t^j - \boldsymbol{\theta}\| \leq \|\boldsymbol{\theta}_t^i - \boldsymbol{\theta}\|_2 \|\boldsymbol{\theta}_t^j - \boldsymbol{\theta}\|_2$$

Finally, we also bound the variance:

$$Var(\boldsymbol{g}_t^i | \hat{\boldsymbol{\theta}}_t, \boldsymbol{c}_t^i, \boldsymbol{c}_{t+}^i) = Var((r_t^i + \gamma (\boldsymbol{\theta}_t^i)^T \boldsymbol{c}_{t+}^i - (\boldsymbol{\theta}_t^i)^T \boldsymbol{c}_t^i) \boldsymbol{c}_t^i | \hat{\boldsymbol{\theta}}_t, \boldsymbol{c}_t^i, \boldsymbol{c}_{t+}^i) = Var(r_t^i \boldsymbol{c}_t^i | \hat{\boldsymbol{\theta}}_t, \boldsymbol{c}_t^i, \boldsymbol{c}_{t+}^i) \leq$$

$$\leq \boldsymbol{c}_t^i Var(r_t^i | \hat{\boldsymbol{\theta}}_t, \boldsymbol{c}_t^i, \boldsymbol{c}_{t+}^i) (\boldsymbol{c}_t^i)^T \leq \boldsymbol{c}_t^i (\boldsymbol{c}_t^i)^T R_{max}^2$$

Hence we have:

$$Tr(\frac{1}{m^2} \sum_{i=1}^{m} Var(\boldsymbol{g}_t^i | \hat{\boldsymbol{\theta}}_t, \boldsymbol{c}_t^i, \boldsymbol{c}_{t+}^i)) \leq \frac{R_{max}^2}{m}$$

Hence we obtain a recursive inequality:

$$\mathbb{E}[\|\boldsymbol{\theta} - \bar{\boldsymbol{\theta}}_{t+1}\|_2^2] \leq \mathbb{E}[\|\boldsymbol{\theta} - \bar{\boldsymbol{\theta}}_t\|_2^2] -$$

$$-\left(\frac{2\alpha_t \omega(1 - \gamma) - \alpha_t^2}{m^2}\right) \mathbb{E}\Big[ \sum_{i=1}^{m} \sum_{j=1}^{m} \|\boldsymbol{\theta} - \boldsymbol{\theta}_t^i\|_2 \|\boldsymbol{\theta} - \boldsymbol{\theta}_t^j\|_2\Big] + \frac{\alpha_t^2}{m} R_{max}^2$$

We can now use a Lemma H.3 to get:

$$\mathbb{E}[\|\boldsymbol{\theta} - \bar{\boldsymbol{\theta}}_{t+1}\|_2^2] \leq \mathbb{E}[\|\boldsymbol{\theta} - \bar{\boldsymbol{\theta}}_t\|_2^2](1 - 2\alpha_t \omega(1 - \gamma) + \alpha_t^2) + \frac{\alpha_t^2}{m} R_{max}^2$$

We now set $\alpha_t = \frac{\beta}{\Lambda + \frac{t}{\omega}}$ with $\beta = \frac{2}{(1-\gamma)\omega}$ and $\Lambda = \frac{16}{(1-\gamma)^2 \omega}$ . We observe that since $\alpha_t \leq (1 - \gamma)\omega$ we have that $-2\alpha_t \omega(1 - \gamma) + \alpha_t^2 R_{max}^2 = \alpha_t \omega(1 - \gamma)(-2 + \frac{\alpha_t R_{max}^2}{\omega(1-\gamma)}) \leq \alpha_t \omega(1 - \gamma)(-2 + 1) = -\alpha_t \omega(1 - \gamma)$. Let us now define

$\nu = \max\{\beta^2 \frac{R_{max}^2}{m}, \Lambda \|\boldsymbol{\theta} - \frac{1}{m}\sum_{i=1}^m \hat{\boldsymbol{\theta}}_0^i\|_2^2\}$ *and observe that we have* $\mathbb{E}[\|\boldsymbol{\theta} - \bar{\boldsymbol{\theta}}_0\|_2^2] \leq \frac{\nu}{\Lambda}$. *Proceeding by induction, let us suppose that* $\mathbb{E}[\|\boldsymbol{\theta} - \bar{\boldsymbol{\theta}}_t\|_2^2] \leq \frac{\nu}{\hat{t}}$ , *where* $\hat{t} = \omega t + \Lambda$. *We then have:*

$$\mathbb{E}[\|\boldsymbol{\theta} - \bar{\boldsymbol{\theta}}_{t+1}\|_2^2] \leq \mathbb{E}[\|\boldsymbol{\theta} - \bar{\boldsymbol{\theta}}_t\|_2^2](1 - \alpha_t \omega(1-\gamma)) + \frac{\alpha_t^2 R_{max}^2}{m}$$

$$\leq \frac{\nu}{\hat{t}}(1 - \frac{\beta\omega(1-\gamma)}{\hat{t}}) + \frac{R_{max}^2 \beta^2}{\hat{t}^2 m} = \frac{\hat{t}-1}{\hat{t}^2}\nu + \frac{\frac{R_{max}^2 \beta^2}{m} - ((1-\gamma)\omega\beta - 1)\nu}{\hat{t}^2}$$

$$= \frac{\hat{t}-1}{\hat{t}^2}\nu + \frac{\frac{R_{max}^2 \beta^2}{m} - \nu}{\hat{t}^2}$$

*We now observe that by definition of* $\nu$ *we have* $\frac{R_{max}^2 \beta^2}{m} \leq \nu$ *and that* $\hat{t}^2 \geq (\hat{t}-1)(\hat{t}+1)$. *Thus we have:*

$$\mathbb{E}[\|\boldsymbol{\theta} - \bar{\boldsymbol{\theta}}_{t+1}\|_2^2] \leq \frac{\nu}{\hat{t}+1} \leq \frac{\nu}{\hat{t}+\frac{1}{\omega}}$$

*By induction this proves that:*

$$\mathbb{E}[\|\boldsymbol{\theta} - \bar{\boldsymbol{\theta}}\|_2^2] \leq O\left(\max\left\{\frac{\max\{\frac{R_{max}^2}{\omega m}, \|\boldsymbol{\theta} - \frac{1}{m}\sum_{i=1}^m \hat{\boldsymbol{\theta}}_0^i\|_2^2\}}{1 + (1-\gamma)^2 n}, CP\right\}\right)$$

*We now use the fact that* $\omega \leq S\pi_{min}\lambda_{min}$ *to get the statement of the Theorem.*

## G    Analysis of worst-case TD initial bias

Given the average initial parameter value $\hat{\boldsymbol{\theta}}_0$, assume we are faced with a problem where the features of states are chosen so that $\hat{\boldsymbol{\theta}}_0$ is an eigenvector of the $X^T X$ matrix with eigenvalue $\lambda_{\min}$. We see that if we form vector $\boldsymbol{u}$ consisting of value function for each state we get $\|u\|_2^2 \leq O(\frac{SR_{\max}^2}{(1-\gamma)^2})$. We choose the true parameter vector $\boldsymbol{\theta}$ to be parallel to $\hat{\boldsymbol{\theta}}_0$, hence $\|X^T\boldsymbol{\theta}\|_2^2 \leq \lambda_{\min}\|\boldsymbol{\theta}\|_2^2 \leq O(\frac{SR_{\max}^2}{(1-\gamma)^2})$. We thus have $\|\boldsymbol{\theta}\|_2^2 \leq O(\frac{SR_{\max}^2}{(1-\gamma)^2\lambda_{\min}})$. We can now set $\boldsymbol{\theta} = -\hat{\boldsymbol{\theta}}_0$ and observe that we have $\|\boldsymbol{\theta} - \hat{\boldsymbol{\theta}}_0\|_2^2 = 4\|\boldsymbol{\theta}\|_2^2 \leq O(\frac{SR_{\max}^2}{(1-\gamma)^2})$.

## H    Lemmas for TD learning

Within this appendix, we propose Lemma H.1, which is a more general version of Lemma 1 and 3 from Bhandari et al. (2018). We also propose Lemma H.3, which is a simple consequence of the triangle inequality. Both of these Lemmas consistute very useful tools for our analysis of distributed TD learning.

**Lemma H.1** *For TD value function approximation problem in a MDP with discount factor* $\gamma$, *with* $\omega$ *being the smallest eigenvalue of the feature matrix weighted by the stationary distribution, for* $g_t^i = (r_t + \gamma \boldsymbol{c}_{t+}^T \hat{\boldsymbol{\theta}}^i - \boldsymbol{c}_t^T \hat{\boldsymbol{\theta}}^i)\boldsymbol{c}_t$ *being the gradient step, we have that:*

$$(\boldsymbol{\theta} - \boldsymbol{\theta}_t^i)^T \mathbb{E}[g_t^j | \hat{\boldsymbol{\theta}}_t^j] \geq \omega(1-\gamma)\|\boldsymbol{\theta} - \boldsymbol{\theta}_t^i\|_2 \|\boldsymbol{\theta} - \boldsymbol{\theta}_t^j\|_2$$

**Proof H.2** *Let us define* $\xi_i = (\boldsymbol{\theta} - \hat{\boldsymbol{\theta}}^i)\boldsymbol{c}_t^i$, $\xi_i' = (\boldsymbol{\theta} - \hat{\boldsymbol{\theta}}^i)\boldsymbol{c}_{t+}^i$ *and* $\xi_{i,j} = (\boldsymbol{\theta} - \hat{\boldsymbol{\theta}}^i)\boldsymbol{c}_t^j$. *Observing that the expected gradient is zero at the optimal value of* $\boldsymbol{\theta}$ *we get:*

$$\mathbb{E}[g_t^j | \hat{\boldsymbol{\theta}}_t^j] = \mathbb{E}[g_t^j | \hat{\boldsymbol{\theta}}_t^j] - \mathbb{E}[(r_t^j + \gamma\boldsymbol{\theta}^T \boldsymbol{c}_{t+}^j - \boldsymbol{\theta}^T \boldsymbol{c}_t^j)\boldsymbol{c}_t | \hat{\boldsymbol{\theta}}_t^j] = \mathbb{E}[\boldsymbol{c}_t^j(\gamma\boldsymbol{c}_{t+}^j - \boldsymbol{c}_t^j)(\hat{\boldsymbol{\theta}}_t^j - \boldsymbol{\theta}) | \hat{\boldsymbol{\theta}}_t^j] =$$

$$= \mathbb{E}[\boldsymbol{c}_t^j(\boldsymbol{c}_t^j - \gamma\boldsymbol{c}_{t+}^j) | \hat{\boldsymbol{\theta}}_t^j]$$

*Hence we get:*

$$(\boldsymbol{\theta} - \hat{\boldsymbol{\theta}}_t^i)^T \mathbb{E}[g_t^j | \hat{\boldsymbol{\theta}}_t^j] = \mathbb{E}[\xi_{i,j}(\xi_j - \gamma\xi_j') | \hat{\boldsymbol{\theta}}_t^j, \hat{\boldsymbol{\theta}}_t^i] = \mathbb{E}[\xi_{i,j}\xi_j | \hat{\boldsymbol{\theta}}_t^j, \hat{\boldsymbol{\theta}}_t^i] - \gamma\mathbb{E}[\xi_{i,j}\xi_j' | \hat{\boldsymbol{\theta}}_t^j, \hat{\boldsymbol{\theta}}_t^i]$$

*Since $\boldsymbol{c}_t^j$ and $\boldsymbol{c}_t^i$ are independent we have that given estimate of parameter vectors $\xi_{i,j}$ and $\xi_j$ are also independent. Moreover by Cauchy-Schwarz we have that*

$$\mathbb{E}[\xi_{i,j}\xi_j'|\hat{\boldsymbol{\theta}}_t^j, \hat{\boldsymbol{\theta}}_t^i] \leq \sqrt{\mathbb{E}[\xi_{i,j}^2|\hat{\boldsymbol{\theta}}_t^j, \hat{\boldsymbol{\theta}}_t^i]\mathbb{E}[(\xi_j')^2|\hat{\boldsymbol{\theta}}_t^j, \hat{\boldsymbol{\theta}}_t^i]}$$

*. We now observe that:*

$$\mathbb{E}[\xi_{i,j}\xi_j|\hat{\boldsymbol{\theta}}_t^j, \hat{\boldsymbol{\theta}}_t^i] = \sum_{s \in S} \pi(s)(\boldsymbol{\theta} - \hat{\boldsymbol{\theta}}_t^i)^T \boldsymbol{c}_t^i (\boldsymbol{\theta} - \hat{\boldsymbol{\theta}}_t^j)^T \boldsymbol{c}_t^i$$

$$= (\boldsymbol{\theta} - \hat{\boldsymbol{\theta}}_t^i)^T \Sigma (\boldsymbol{\theta} - \hat{\boldsymbol{\theta}}_t^j) \leq (\boldsymbol{\theta} - \hat{\boldsymbol{\theta}}_t^i)^T (\boldsymbol{\theta} - \hat{\boldsymbol{\theta}}_t^j)\omega \leq \omega\|\boldsymbol{\theta} - \hat{\boldsymbol{\theta}}_t^i\|_2 \|\boldsymbol{\theta} - \hat{\boldsymbol{\theta}}_t^j\|_2$$

*Moreover by similar argument we can show that:* $\mathbb{E}[\xi_{i,j}^2|\hat{\boldsymbol{\theta}}_t^j, \hat{\boldsymbol{\theta}}_t^i] \leq \omega\|\boldsymbol{\theta} - \hat{\boldsymbol{\theta}}_t^i\|_2^2$ *and* $\mathbb{E}[\xi_j^2|\hat{\boldsymbol{\theta}}_t^j, \hat{\boldsymbol{\theta}}_t^i] \leq \omega\|\boldsymbol{\theta} - \hat{\boldsymbol{\theta}}_t^j\|_2^2$ *which complete the proof.*

**Lemma H.3** $\|\boldsymbol{\theta} - \bar{\boldsymbol{\theta}}_t\|_2^2 \leq \frac{1}{m^2} \sum_{i=1}^m \|\boldsymbol{\theta} - \hat{\boldsymbol{\theta}}_t^i\|_2 \sum_{j=1}^m \|\boldsymbol{\theta} - \hat{\boldsymbol{\theta}}_t^j\|_2$

**Proof H.4**

$$\|\boldsymbol{\theta} - \bar{\boldsymbol{\theta}}_t\|_2^2 = \|\boldsymbol{\theta} - \frac{1}{m}\sum_{i=1}^m \hat{\boldsymbol{\theta}}_t^i\|_2^2 = \left(\|\frac{1}{m}\sum_{i=1}^m (\boldsymbol{\theta} - \hat{\boldsymbol{\theta}}_t^i)\|_2\right)^2 = \frac{1}{m^2}\left(\|\sum_{i=1}^m (\boldsymbol{\theta} - \hat{\boldsymbol{\theta}}_t^i)\|_2\right)^2$$

*By triangle inequality we have:*

$$\|\boldsymbol{\theta} - \bar{\boldsymbol{\theta}}_t\|_2^2 \leq \frac{1}{m^2}\left(\sum_{i=1}^m \|(\boldsymbol{\theta} - \hat{\boldsymbol{\theta}}_t^i)\|_2\right)^2 = \frac{1}{m^2}\sum_{i=1}^m \sum_{j=1}^m \|(\boldsymbol{\theta} - \hat{\boldsymbol{\theta}}_t^i)\|_2 \|(\boldsymbol{\theta} - \hat{\boldsymbol{\theta}}_t^j)\|_2$$

