# OpenReview forum: "Settling the Communication Complexity for Distributed Offline Reinforcement Learning"
_TMLR — Rejected by TMLR_

### Review · Reviewer_YR4R · 2022-12-11

**Summary Of Contributions:**

This paper studies the problem of value function estimation for contextual bandits and reinforcement learning (RL) with multiple parallel machines, under constraints on their communication complexity. The particular problem is to estimate the value function of a given policy under distributed data obtained by the same policy, with a realizability assumption in a linear function class. The paper derives mostly matching lower and upper bounds for this problem, where upper bounds are obtained by distributed least-squares style algorithms.

**Audience:**

Yes

**Claims And Evidence:**

Yes

**Requested Changes:**

* Some results where behavior policy is different from target policy could make the results substantially stronger and more aligned with current offline RL literature.

* The discussion about related work needs to be substantially expanded to make the paper into a publishable form.

* Could the authors explain how technically the results are different from prior results on linear regression with communication complexity constraints, e.g. of Zhang et al. (2013) or other existing work along this line?


**Strengths And Weaknesses:**

Strengths:

* The problem of distributed estimation with communication constraints is well studied in various ML/stats problems in general, but seems rarely studied in contextual bandits / RL settings. Obtaining results in this setting would definitely be of interest.

* The established upper bounds mostly match the lower bounds.


Weaknesses:

* While the paper talks about “Offline RL”, its actual problem setting seems much easier than standard offline bandit/RL settings. For example, the RL setting studied in this paper is to estimate $V^{\pi}$ from data obtained also from $\pi$, which is a standard linear regression-like problem without any distribution shift. In comparison, the Offline Policy Evaluation problem (a widely studied problem in offline RL) typically assumes the data generating policy (the “behavior policy”) is different from the target policy $\pi$, see e.g. [1-2] and the many references therein. In my opinion, the setting in this paper is just canonical statistical estimation, and does not capture the real difficulty of bandits/RL.
[1] Xie, T., Ma, Y., & Wang, Y. X. (2019). Towards optimal off-policy evaluation for reinforcement learning with marginalized importance sampling. Advances in Neural Information Processing Systems, 32.
[2] Yin, M., Bai, Y., & Wang, Y. X. (2021, March). Near-optimal provable uniform convergence in offline policy evaluation for reinforcement learning. In International Conference on Artificial Intelligence and Statistics (pp. 1567-1575). PMLR.

* Due to the above direct similarity to standard linear regression problem under communication constraints, technically, I suspect the results should mostly be straightforward adaptations of the results in Zhang et al. (2013) (e.g. their Section 4). I understand to apply their results in the very setting here may require handling the problem structure of bandits/MDPs correctly (to pose it as linear regression), but that part sounds standard in my opinion.

* There is a substantial lack of related work discussions in this paper (for one thing, only 6 papers are cited in the reference). In particular the paper does not seem to discuss the vast recent literature on offline RL (for both the Offline Policy Evaluation problem and also the Offline Learning problem, e.g. [3-4] and the references therein). In its current form, it is quite hard to situate the results within the literature.
[3] Chen, J., & Jiang, N. (2019, May). Information-theoretic considerations in batch reinforcement learning. In International Conference on Machine Learning (pp. 1042-1051). PMLR.
[4] Jin, Y., Yang, Z., & Wang, Z. (2021, July). Is pessimism provably efficient for offline rl?. In International Conference on Machine Learning (pp. 5084-5096). PMLR.

---

> ### Author Response · Authors · 2023-01-05
> **Response to Reviewer YR4R**
>
> We want to thank the reviewer for their time spent reading our work and their valuable feedback.
> Regarding the mentioned weaknesses:
> > While the paper talks about “Offline RL”, its actual problem setting seems much easier than standard offline bandit/RL settings. For example, the RL setting studied in this paper is to estimate  from data obtained also from , which is a standard linear regression-like problem without any distribution shift. In comparison, the Offline Policy Evaluation problem (a widely studied problem in offline RL) typically assumes the data generating policy (the “behavior policy”) is different from the target policy , see e.g. [1-2] and the many references therein. In my opinion, the setting in this paper is just canonical statistical estimation, and does not capture the real difficulty of bandits/RL.
> [1] Xie, T., Ma, Y., & Wang, Y. X. (2019). Towards optimal off-policy evaluation for reinforcement learning with marginalized importance sampling. Advances in Neural Information Processing Systems, 32.
> [2] Yin, M., Bai, Y., & Wang, Y. X. (2021, March). Near-optimal provable uniform convergence in offline policy evaluation for reinforcement learning. In International Conference on Artificial Intelligence and Statistics (pp. 1567-1575). PMLR.
>
> Our work deals with the parameter identification problem, which poses different challenges than standard regret minimization. While it is true that we essentially reduce the problems to statistical inference problems, the reduction itself is a novel idea that allows to leverage powerful results from vast literature on distributed statistical inference.
>
> > Due to the above direct similarity to standard linear regression problem under communication constraints, technically, I suspect the results should mostly be straightforward adaptations of the results in Zhang et al. (2013) (e.g. their Section 4). I understand to apply their results in the very setting here may require handling the problem structure of bandits/MDPs correctly (to pose it as linear regression), but that part sounds standard in my opinion.
>
> While it is true that our work is based on the seminal paper by Zhang et al. (2013), our results require a novel reduction of the MDP to a statistical inference problem.
>
> > There is a substantial lack of related work discussions in this paper (for one thing, only 6 papers are cited in the reference). In particular the paper does not seem to discuss the vast recent literature on offline RL (for both the Offline Policy Evaluation problem and also the Offline Learning problem, e.g. [3-4] and the references therein). In its current form, it is quite hard to situate the results within the literature.
> [3] Chen, J., & Jiang, N. (2019, May). Information-theoretic considerations in batch reinforcement learning. In International Conference on Machine Learning (pp. 1042-1051). PMLR.
> [4] Jin, Y., Yang, Z., & Wang, Z. (2021, July). Is pessimism provably efficient for offline rl?. In International Conference on Machine Learning (pp. 5084-5096). PMLR.
>
> We thank the reviewer for pointing us towards relevant related work. We agree that the related work section could be expanded and we will rewrite it.

---

### Review · Reviewer_jeD8 · 2022-12-13

**Summary Of Contributions:**

This paper studies a distributed parameter estimation problem for contextual bandits, episodic MDPs, and non-episodic MDPs parameterized with linear models. m machines will collect data independently, and send a single message to the center. The paper derives lower bounds for the estimation error in terms of communication budget (in bits), number of actions, feature dimension, number of machines, number of samples, etc. For the contextual bandit and episodic MDP case, matching upper and lower bounds are derived; for the non-episodic setting, there is a gap that is governed by the initial error.

**Audience:**

Yes

**Claims And Evidence:**

Yes

**Requested Changes:**

- Please make the proof sketches in the main text self-contained so that readers do not need to refer to the formal proof.
- Please make it clear in the abstract and intro that the hardness of the non-episodic setting is not formally proven through a lower bound, and that there is still gap between upper/lower bounds.

**Strengths And Weaknesses:**

Strength
- The problem itself is important, and the paper provides a good characterization for the problem.
- The paper identifies a difference between the episodic and non-episodic settings in terms of algorithm design.

Weakness
- The contribution might be a bit marginal given the work of Zhang et al. (2013), since they already derive communication lower bound for distribution optimization. This work is mostly an application of their result to the linear setting.
- For the non-episodic setting, there is no formal proof showing that it is indeed harder than the episodic setting. The hardness arguments only come from explaining the difficulty of algorithm design.
- The difficulty of exploration in RL is assumed away by the assumption that the data is well-conditioned.
- In terms of writing, the proof sketches do not work well for understanding the idea of the proof. For example, in Proof 3.4, the sketch constantly refer to some quantity that is only seen in the appendix (e.g., v_{j,k}, "second and third term in Lemma A.4"). It's thus not possible to understand the proof sketch without looking at the real proof.
- I don't understand why the number of actions A appeared in the bounds for contextual bandits, while not appeared in MDPs.

Overall, the contribution is a bit marginal given the work of Zhang et al. (2013), and for the most interesting part of non-episodic MDP where the authors claim to be strictly more difficult, there is no formal proof on the lower bound.

---

> ### Author Response · Authors · 2023-01-05
> **Response to Reviewer jeD8**
>
> We would like to thank the reviewer for their time and insightful suggestions.
> Regarding the mentioned weaknesses
>
> > The contribution might be a bit marginal given the work of Zhang et al. (2013), since they already derive communication lower bound for distribution optimization. This work is mostly an application of their result to the linear setting.
>
> While it is true, we utilise the seminal work of Zhang et al. (2013), our work presents a novel reduction of offline RL problems to statistical inference problems.
>
> > For the non-episodic setting, there is no formal proof showing that it is indeed harder than the episodic setting. The hardness arguments only come from explaining the difficulty of algorithm design
>
> The increase in difficulty comes from the fact that in Theorem 7 the denominator is multiplied by another $
> \pi_{max} (1- \gamma)^2$ factor (which must be smaller than 1) compared to the bound in Theorem 5.
>
> > The difficulty of exploration in RL is assumed away by the assumption that the data is well-conditioned.
>
> As we work in the offline setting, we assume the data has already been generated and we have no impact on how this was done. Hence, the difficulty of the problem boils down to how well-conditioned the collected data is.
>
> > In terms of writing, the proof sketches do not work well for understanding the idea of the proof. For example, in Proof 3.4, the sketch constantly refer to some quantity that is only seen in the appendix (e.g., v_{j,k}, "second and third term in Lemma A.4"). It's thus not possible to understand the proof sketch without looking at the real proof.
>
> We thank the reviewer for pointing out the omission of not defining v_{j,k} in the main body. When it comes to referring to the Lemmas from Appendix, we will add a short one-sentence description of what each Lemma is about in the proof sketch.
>
> > I don't understand why the number of actions A appeared in the bounds for contextual bandits, while not appeared in MDPs.
>
> This is because the MDP problem concerns with estimating the parameter mapping state features to value function for a given behaviour policy. Hence the problem is not neccessarily geting more difficult as we have more actions. In the bandit case, we studied the identification of parameter for each arm, hence it naturally had to scale with the number of arms (actions).

---

### Review · Reviewer_yTuA · 2022-12-23

**Summary Of Contributions:**

This paper studies distributed policy evaluation in batch reinforcement learning. In the setting of this paper, data is collected by M machines that, after computing local estimates, can send a single message of B bits to the global server, which outputs the global estimate. The motivation is a IoT scenario where wearable devices collect a lot of real-time interaction data and communication with the server is rare in comparison.

First the setting of reward estimation in linear contextual bandits is considered. Here a lower bound on the worst-case risk (defined in terms of MSE) is established that is inversely proportional to the number of machines once we assume each machine collects a fixed amount of data for each action. The lower bound prescribes an optimal communication budget B = Ad, where A is the number of actions and d is the dimension of context vectors. This lower bound is matched up to logarithmic terms by a simple algorithm: each machine computes a least-squares estimate of the reward parameter and sends it (with finite precision) to the central server; the output of the server is the average of the M estimates.

The same approach is used for policy evaluation in finite-horizon MDPs, assuming that the state-value function for the given policy is linear and every state is visited n times by each machine. The lower bound on the risk is again inversely proportional to the number of machines (with a square dependence on the task horizon), and is matched by an averaged Monte Carlo evaluation algorithm.
Moving to infinite-horizon discounted MPDs, the lower bound now scales as $(1-\gamma)^{-2}$, where $\gamma$ is the discount factor. Here MC is not an option. The proposed algorithm is semi-gradient TD with final averaging of the parameters in the central server. In this case there is a bias term that cannot be reduced by increasing the number of machines.

**Audience:**

Yes

**Broader Impact Concerns:**

I don't think a Broader Impact Statement is needed for this kind of work.

**Claims And Evidence:**

No

**Requested Changes:**

Critical proposed changes:

Under the current assumptions, the results on bandits and RL may appear as a trivial extension of similar results for supervised learning (Zhang et al. 2013) that do not capture the fundamental aspects of interactive decision making.
So, although accurate, the claims made in the paper are not convincing as claims on learning in linear bandits and MDPs.
However, I do not think this is necessarily the case, especially given the effort in designing hard instances for these problems. I think that the results of this paper can be made more meaningful just by a more careful modeling of the bandit and RL problems. Here are, in more detail, my suggestions on how to modify the assumptions:

c1. Contextual bandits: first of all, it should be stated more clearly whether the contexts are stochastic or adversarial. Since the lower bound (Thm 3.3) has a clear adversarial flavor (you set the values of the context vectors as needed), I suggest you go for the adversarial setting. Then, the statement "eigenvalues of the covariance matrix naturally grow with the number of samples" is not meaningful and Assumption 3.2. should be motivated in another way (e.g., by elaborating on the "unidentifiability" problem) or replaced by a weaker assumption.

c2. As for the assumption that each agent has collected n samples for each action, it is not meaningful. You should specify whether all the agents are playing the same policy (and how: are they shipped with the same policy or do they need to communicate?) or different policies with similar properties. Then, you should make assumptions on the coverage of this policy (or policies), for instance by assuming a minimum probability of selecting each action (a "forced-exploration" component). From this, the number of samples for each action can be established in expectation or in probability. You should also clarify whether the policies run by the agents are specifically designed to solve the reward estimation problem or not. Notice that there is a whole literature on optimal experimental design and best-arm identification.

c3. Linear MDPs: I suggest you start from a common assumption such as low-rank MDP (Jin et al. 2020). In this case, for any policy, $Q^\pi(s,a)=\phi(s,a)^T\theta^\pi$. Then you could argue that, for a fixed policy, $V^\pi(s)=\phi(s)^T\theta^\pi$, where $\phi(s)=E[\phi(s,a)]$, and motivate the fact that the agent is able to compute these state features; alternatively, you could consider the problem of estimating the Q function instead of the V function.

c4. Assumption 4.2 is too artificial. Again, you can make the setting more meaningful by modeling the agents as running a fixed policy (here they must all run the same policy for the policy evaluation problem to make sense), make assumptions on the policy and from these derive the number of samples per state (in expectation or in probability). The assumption of the policy could be in terms of coverage of the state space (e.g., minimum probability of visiting any state). However, in the setting of linear function approximation, defining the coverage directly in feature space would be more meaningful, and may avoid the counterintuitive 1/S factor in the MSE. This could be an assumption on the minimum eigenvalue of the feature covariance matrix (similarly to the linear bandit setting), where actions are selected according to the fixed policy of the agent. In the context of offline RL, a similar assumption has been used for instance in "Infinite-Horizon Offline Reinforcement Learning with Linear Function Approximation: Curse of Dimensionality and Algorithm" (Chen et al. 2021).

Minor (non-critical) suggested changes:

- Table 1, "card. of state space": unnecessary abbreviation
- I suggest to replace "gameplay" with "interaction"
- Sec. 3 "the agent is faced with a number of slot machines": this is just an example and should not be used as the definition of the problem
- To be useful, the proof sketches should be self-contained. Currently, several quantities and statements that appear in the sketches are only defined in the appendix (Lemma A.4, I(V,Y), $\delta$, $v_{j,k}$, $Q_{j,k}$
- In section 4 $G_s$ as defined also depends on $t$ so it should be $G_{s,t}$
- $p$ is a confusing notation for a Bernoulli random variable since it usually denotes the parameter of the Bernoulli distribution
- Proof 5.3 "states in the same state"
- Proof 5.3 "consistency equation" do you mean Bellman's equation?
- The sequential numbering of theorems and proofs is confusing since it is not clear, for instance, that "Proof A.3" is the proof of Lemma A.2 and not the proof of a "Lemma A.3"
- In Proof A.3, it is not clear why the first inequality holds for any i, and not just for the Y_i with the largest mutual information
- The last equality of Proof A.3 is also not clear
- In Proof B.1, shouldn't boldface V be the concatenation of A vectors (one for each arm)?
- In Proof B.1, setting B_{j,k} to an event with probability 1 seems a bit redundant, and may suggest that a more direct argument is possible.
- In Proof E.1, I believe an $r_a$ is missing from the third equation on page 20

**Strengths And Weaknesses:**

The paper studies the problem of communication in interactive learning in a difficult setting (single communication round) and using a meaningful metric (the number of bits used to transmit estimates with finite precision).
The different settings (contextual bandits, finite-horizon MDPs, discounted MDPs) are analyzed in a complete way by first providing a lower bound on the MSE based on a hard instance, then an algorithm with a (matching, in the first two cases) upper bound.
The paper is clear in its exposition of the problem, the results, and the related literature, and is well organized.

The main weakness of this work lies in the assumptions on the interaction protocol. Given how the results are derived by adapting existing results in statistical learning, I understand the need of framing bandit and RL problems in a way that can be analyzed with this tools. However, the resulting assumptions are very strong, and not really representative of the peculiar aspects of interactive learning systems.
More specifically:

w1. Contextual bandits: the assumption that each agent has collected n samples for each arm is not a natural one for this setting. First, the fact that all agents are using the same policy should be motivated, if this is the case. Then, even considering a uniform policy, the number of samples per action would be N/A only in expectation. Alternatively, you could state that the samples for each arm are at least n with high probability after a large enough number of rounds. However, unless the agents are playing a uniform or round-robin policy, many samples from the arms that are pulled more often would be wasted in your formulation.

w2. Contextual bandits: it is not clear whether the contexts are stochastic (sampled from a fixed distribution) or adversarial (selected by an adversary). In the adversarial case, Assumption 3.2. is a limitation on the power of the adversary that should be motivated further, as the statement "eigenvalues of the covariance matrix naturally grow with the number of samples" is not meaningful in this scenario. In the stochastic setting, this informal statement can be made formal as follows: instead of assuming X^TX has a minimum eigenvalue that grows linearly with n, you can assume that the covariance matrix E[cc^T], where c is sampled from a fixed distribution, has min eigenvalue $\lambda$. Then you can use Matrix Azuma (User-friendly tail bounds for sums of random matrices, Tropp, 2012) to show that, with high probability, X^TX has min eigenvalue lower bounded by $n\lambda - o(n)$.

w3. Episodic MDPs: requiring the state-value function to be linear in state features is not a common assumptions in RL. The most common assumption for linear function approximation is that of low-rank MDPs (Jin et al. 2020) which implies the Q (state-action) value function is linear in state-action vectors.

w4. Assumption 4.2: this assumption is very strong (as acknowledged by the authors) and should be motivated further. This is reflected by the risk being inversely proportional to the number of states, which suggests that this bound is not very meaningful. A more rigorous alternative to the informal statement "each new state effectively increases the number of available samples for the same number of features" would be to characterize the coverage in feature space rather than in state space (more details below). Similar concerns apply to the discounted setting.

---

> ### Author Response · Authors · 2023-01-05
> **Response to Reviewer yTuA**
>
> We would like to thank the reviewer for their time and insightful comments and suggestions.
> Regarding the mentioned weaknesses:
>
> w1:
> > Contextual bandits: the assumption that each agent has collected n samples for each arm is not a natural one for this setting. First, the fact that all agents are using the same policy should be motivated, if this is the case. Then, even considering a uniform policy, the number of samples per action would be N/A only in expectation. Alternatively, you could state that the samples for each arm are at least n with high probability after a large enough number of rounds. However, unless the agents are playing a uniform or round-robin policy, many samples from the arms that are pulled more often would be wasted in your formulation.
>
> We agree that the assumption that we have at least n sample for each arm can be a bit limiting, but the parameter identification setting in which we operate required such an assumption to be made. If there is at least one arm, which has been pulled significantly less number of times than others, then the expected error in estimating its parameters will quickly dominate the errors coming from parameters corresponding to other arms.
>
> w2:
> > Contextual bandits: it is not clear whether the contexts are stochastic (sampled from a fixed distribution) or adversarial (selected by an adversary). In the adversarial case, Assumption 3.2. is a limitation on the power of the adversary that should be motivated further, as the statement “eigenvalues of the covariance matrix naturally grow with the number of samples” is not meaningful in this scenario. In the stochastic setting, this informal statement can be made formal as follows: instead of assuming X^TX has a minimum eigenvalue that grows linearly with n, you can assume that the covariance matrix E[cc^T], where c is sampled from a fixed distribution, has min eigenvalue . Then you can use Matrix Azuma (User-friendly tail bounds for sums of random matrices, Tropp, 2012) to show that, with high probability, X^TX has min eigenvalue lower bounded by $n \lambda - o(n)$
>
> The contexts are stochastic. We bounded the eigenvalues, as otherwise, a large change in the parameter of interest would produce a small change in the expected reward. We thank the reviewer for meaningful suggestions regarding the bounds of the expected eigenvalues. These will definitely be useful for extending our work.
>
> w3:
> > Episodic MDPs: requiring the state-value function to be linear in state features is not a common assumptions in RL. The most common assumption for linear function approximation is that of low-rank MDPs (Jin et al. 2020) which implies the Q (state-action) value function is linear in state-action vectors.
>
> We wouldn't agree that the assumption of linear state value function is very uncommon, for example, it has been used in the seminal paper by Bhandari et al. "A finite time analysis of temporal difference learning with linear function approximation."(2018). However, if for some reason we wish to replace this assumption with the assumption of the linear state-action value function, we can indeed follow the argument described by the reviewer (in c), where $\mathbb{E}_{a \sim \pi(s)}[\phi(s,a)]$ is computable as long as we know the agent's policy $\pi(s)$. We would like to thank the reviewer for this insightful suggestion, we will add a discussion on these possible alternative assumptions to the paper.
>
> w4:
> > Assumption 4.2: this assumption is very strong (as acknowledged by the authors) and should be motivated further. This is reflected by the risk being inversely proportional to the number of states, which suggests that this bound is not very meaningful. A more rigorous alternative to the informal statement “each new state effectively increases the number of available samples for the same number of features” would be to characterize the coverage in feature space rather than in state space (more details below). Similar concerns apply to the discounted setting
>
> We thank the reviewer for the meaningful suggestions. Indeed, characterising the convergence in the feature space might prove to produce more intuitive results and constitutes an exciting direction for future work.

---

### Decision · Action_Editors · 2023-02-13

**Recommendation:** Reject

**Comment:**

All 3 reviewers independently claimed that the paper under review is considered to be lacking in terms of its ability to capture the challenges of offline reinforcement learning (RL). The authors' response to the reviews has not alleviated the concerns of them. The setting in which the solution is proposed is deemed too simple and does not accurately reflect the difficulties of offline RL, as seen in recent work on the topic. Additionally, the paper has been deemed to be an direct application of results from communication complexity of statistical estimation, as seen in Zhang et al. (2013).  The authors' answer does not show a concrete intent to fix these issue within the current submission. In light of these issues, the AE recommends rejection of the paper.

**Audience:**

The reviewers agreed that the execution of the paper may need to be significantly improved (e.g. include more related work) before getting published and having the TMLR's audience benefit in knowing the findings of this paper.

**Claims And Evidence:**

All claims and evidence seem correct, however the reviewer find that the peculiar challenges of bandits/RL are ruled out by the strong assumptions made by the authors.